# Construction of an lncRNA-mediated ceRNA network to investigate the inflammatory regulatory mechanisms of ischemic stroke

**Meimei Xu**[1], **Shan Yuan**[1], **Xing Luo**[1], **Mengsi Xu**[2], **Guangze Hu**[1], **Zhe He**[1], **Xinyuan Yang**[1], **Rui Gao**[ID][1,3]*

**1** Department of Biochemistry, College of Medicine, Shihezi University, Shihezi, Xinjiang, China, **2** State Key Laboratory of Sheep Genetic Improvement and Healthy Production, Xinjiang Academy of Agricultural and Reclamation Sciences, Shihezi, Xinjiang, China, **3** College of Coastal Agricultural Sciences, Guangdong Ocean University, Zhanjiang, Guangdong, China

* 18609934189@163.com

**Data Availability Statement:** The mRNA and lncRNA sequencing datasets are publicly available at the Gene Expression Omnibus (GEO) database

## Abstract

Long non-coding RNAs (lncRNAs) are among the most abundant types of non-coding RNAs in the genome and exhibit particularly high expression levels in the brain, where they play crucial roles in various neurophysiological and neuropathological processes. Although ischemic stroke is a complex multifactorial disease, the involvement of brain-derived lncRNAs in its intricate regulatory networks remains inadequately understood. In this study, we established a cerebral ischemia-reperfusion injury model using middle cerebral artery occlusion (MCAO) in male Sprague-Dawley rats. High-throughput sequencing was performed to profile the expression of cortical lncRNAs post-stroke, with subsequent validation using RT-PCR and qRT-PCR. Among the 31,183 lncRNAs detected in the rat cerebral cortex, 551 were differentially expressed between the MCAO and sham-operated groups in the ipsilateral cortex (fold change ≥2.0, P < 0.05). An integrated analysis of the 20 most abundant and significantly differentially expressed lncRNAs (DELs) identified 25 core cytoplasmic DELs, which were used to construct an interaction network based on their targeting relationships. This led to the establishment of a comprehensive lncRNA-miRNA-mRNA regulatory network comprising 12 lncRNAs, 16 sponge miRNAs, and 191 target mRNAs. Gene Ontology (GO) and Kyoto Encyclopedia of Genes and Genomes (KEGG) pathway analyses revealed that differentially expressed mRNAs (DEmRNAs) were significantly enriched in stroke-related pathways. Our analysis predicted four key lncRNAs, four miRNAs, and eleven crucial mRNAs involved in post-transcriptional regulation through competing endogenous RNA (ceRNA) mechanisms. These molecules were shown to participate extensively in post-stroke processes, including angiogenesis, axonal regeneration, inflammatory responses, microglial activation, blood-brain barrier (BBB) disruption, apoptosis, autophagy, ferroptosis, and thrombocytopenia. These findings highlight the role of lncRNAs as multi-level regulators in the complex network of post-stroke mechanisms, providing novel insights into the pathophysiological processes of stroke.

under accession number GSE227317 (https://www.ncbi.nlm.nih.gov/geo/query/acc.cgi?acc=GSE227317). The miRNA sequencing dataset is available at the NCBI Sequence Read Archive (SRA) under BioProject accession PRJNA1149933 (https://dataview.ncbi.nlm.nih.gov/object/PRJNA1149933).

**Funding:** This research was funded by the National Natural Science Foundation of China (Grant No. 32260142) [Website: http://www.nsfc.gov.cn] and the Bingtuan Financial Science and Technology Program (Grant No. 2021BB002) [Website: http://www.bingtuan.gov.cn]. The author who received funding is Rui Gao (G.R.). The funders had no role in study design, data collection and analysis, decision to publish, or preparation of the manuscript.

**Competing interests:** The authors declare no conflicts of interest.

**Abbreviations:** lncRNA, Long noncoding RNA; miRNA, microRNA; mRNA, messenger RNA; DELs, Differentially Expressed LncRNAs; DEMs, Differentially Expressed microRNA; DEGs, Differentially Expressed Genes; DEmRNAs, Differentially Expressed mRNAs; ceRNAs, competing endogenous RNAs; GO, Gene Ontology; KEGG, Kyoto Encyclopedia of Genes and Genomes; BBB, blood–brain barrier; IS, Ischemic stroke; NF-κB, nuclear factor-κB; FDR, false discovery rate; FC, fold change; BP, biological process; CC, cellular component; MF, molecular function; MCAO/, R Middle cerebral artery occlusion/reperfusion; MCAO 2 h/, R 24 h MCAO followed by 24 h of reperfusion; OGD/, R oxygen–glucose deprivation/reperfusion.

## 1 Introduction

Stroke is a leading cause of death and permanent disability worldwide [1]. The latest Global Burden of Disease (GBD) study [2] indicated that stroke is the leading cause of death in the Chinese population, with an overall lifetime risk of stroke in China of 39.9%, ranking first worldwide. Ischemic stroke (IS) is one of the two major subtypes of stroke, accounting for approximately 80–85% of all stroke subtypes [3]. IS is caused by sudden cessation of local blood flow in a supplying artery to the brain. Following an IS attack, ischemic brain tissue undergoes a series of harmful cascade events, including the accumulation of reactive oxygen species, infiltration of immune cells, breakdown of the BBB, and irreversible necrosis of neurons [4]. Currently, stroke diagnosis relies primarily on clinical symptoms and medical imaging technology. The most effective treatments for IS include intravenous thrombolysis within the "time window" [5] and intravascular interventional therapy [6]. However, the therapeutic time window is narrow, and most patients do not exhibit typical imaging changes over time. Missing this therapeutic window may lead to irreversible brain damage [7, 8]. Therefore, identifying the molecular mechanisms of IS pathogenesis, screening effective and reliable biomarkers for the early diagnosis of IS, and providing timely and accurate interventions for IS patients are essential to reduce the incidence and improve the prognosis.

LncRNAs are defined as transcripts longer than 200 nucleotides that lack the ability to encode functional proteins [9]. Once considered "junk" in the mammalian genome, lncRNAs are recognized as transcriptional or post-transcriptional regulators. In transcriptional regulation, lncRNAs bind to chromosomes and modify gene expression [10]. In post-transcriptional regulation, lncRNAs can directly impact gene expression by regulating gene degradation or function as competing endogenous RNAs (ceRNAs) that sponge specific miRNAs, thereby indirectly regulating gene expression [11]. Increasing evidence suggests that many lncRNAs act as ceRNAs, affecting various diseases, such as cancer. For example, in prostate cancer cells, the lncRNA OGFRP1 upregulates SARM1 expression by binding with miR-124-3p, thereby promoting proliferation and inducing apoptosis [12]. Ischemic neurons are sensitive to the aberrant expression of noncoding RNAs, which affect apoptosis, autophagy, proliferation, inflammation, and angiogenesis [3]. Therefore, exploring potential biomarkers and therapeutic targets may provide new perspectives for the treatment and diagnosis of IS.

To investigate the roles of lncRNAs, miRNAs, and mRNAs in cerebral ischemia-reperfusion injury, we employed the MCAO model in rats and analyzed differential gene expression profiles (lncRNAs, miRNAs, and mRNAs) in the MCAO and sham groups at 24 hours post-ischemia using high-throughput sequencing. We determined the subcellular localization of lncRNAs and predicted miRNA targets of DELs using bioinformatics tools. Specifically, we identified potential miRNA binding sites within lncRNAs with miRDB and assessed miRNA-mRNA interactions with miRanda and TargetScan. Gene Ontology (GO) and Kyoto Encyclopedia of Genes and Genomes (KEGG) pathway analyses were conducted to elucidate the biological processes and signaling pathways associated with differentially expressed mRNAs. Additionally, we constructed an lncRNA-miRNA-mRNA regulatory network involved in microglial activation, apoptosis, angiogenesis, and inflammatory responses following stroke. The constructed lncRNA-miRNA-mRNA regulatory network is illustrated in Fig 1. The expression of key network components was validated using reverse transcription-polymerase chain reaction (RT-PCR) and quantitative PCR (qPCR). Our findings offer a comprehensive analysis of lncRNAs in stroke pathogenesis, highlighting the ceRNA regulatory network and its functional implications, and provide novel insights into potential therapeutic strategies for stroke prevention and treatment.

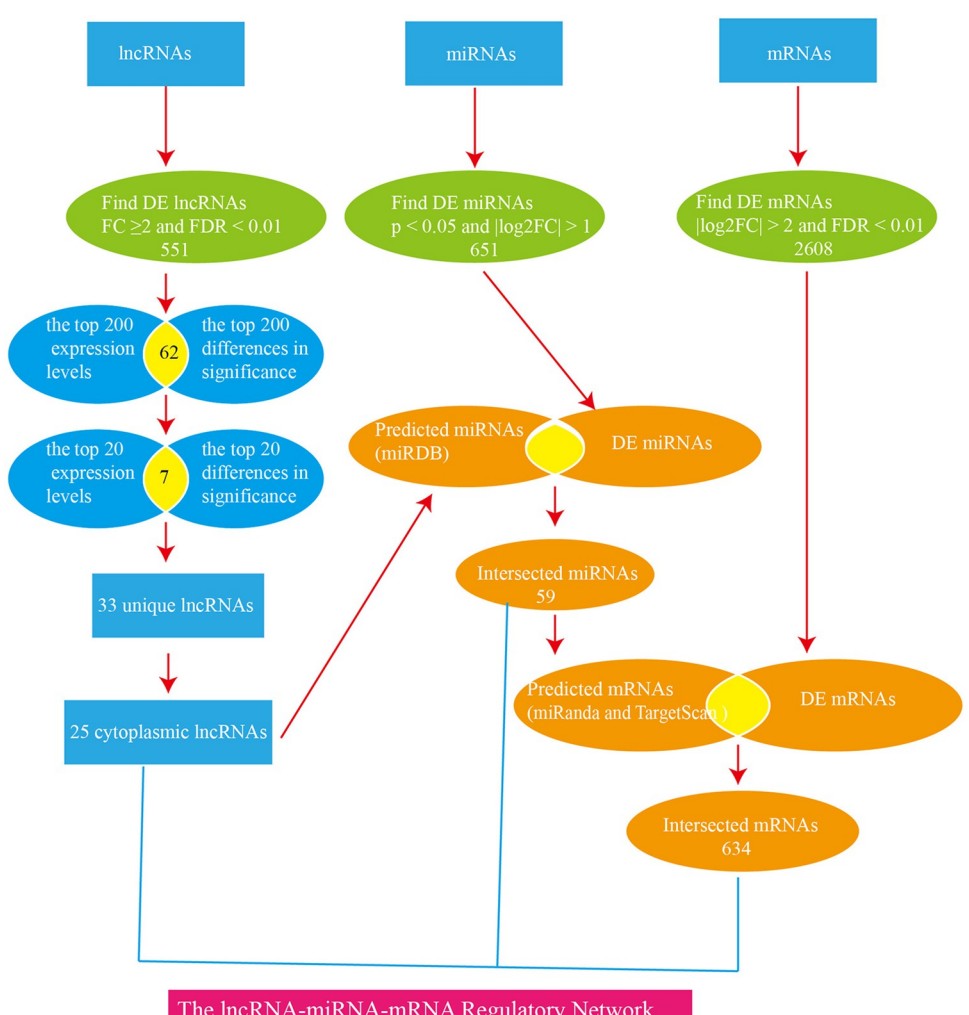

**Fig 1. Construction of long non-coding RNA-mediated competing endogenous RNA network to investigate ischemic stroke.**

## 2 Materials and methods

### 2.1 Animals

Male Sprague–Dawley rats aged eight to ten weeks (220–300 g) were utilized in all the experiments. Twenty-two adult male Sprague–Dawley rats were randomly divided into the MCAO group and the sham-operated group (n = 11 each). In each group, 5 rats were randomly selected for RNA sequencing, 3 for TTC staining to evaluate infarct volume, and 3 for RT–qPCR analysis. The rats used in this study were obtained from the Experimental Animal Center of the Xinjiang Center for Disease Prevention and Control. They were subsequently housed in the specific pathogen-free (SPF) rat facility at the Xinjiang Academy of Agricultural Sciences. Throughout the study, these rats had free access to food and water. The experiments were conducted following the Ethics Guidelines for Animal Experiments of the First Affiliated Hospital, School Medicine of Shihezi University (Xinjiang Province, China).

## 2.2 Middle cerebral artery occlusion/reperfusion (MCAO/R)

The rats were anesthetized using isoflurane inhalation. Specifically, isoflurane was administered at a concentration of 2% through a vaporizer, with a fresh gas flow rate of 4 L/min. The anesthetic gas was delivered to the rats via a nose cone, ensuring a controlled flow rate of 0.41 ml/min. Anesthesia was maintained until the rats exhibited a loss of reflexes, such as the absence of a pinch-paw reflex, confirming an adequate depth of anesthesia for subsequent experimental procedures. Throughout the anesthesia process, vital signs were closely monitored to ensure the animals remained in a pain-free state. The rats were secured and then incised along the midline of the neck to expose the right common carotid artery (CCA), external carotid artery (ECA), and internal carotid artery (ICA). We then ligated the ECA and inserted 4–0 silk threads (Xi-Nong Company, China) into the ICA (18 ± 2 mm) until slight resistance was reached. Two hours after blood flow occlusion, the nylon suture was removed, and reperfusion was performed for 24 h. Sham-operated rats underwent a similar procedure without occlusion of the middle cerebral artery (MCA). At 24 h postreperfusion, the neurological deficits of the rats were assessed following the method described by Longa et al. [13]. The study included rats with scores ranging from 1 to 3.To minimize animal suffering and comply with ethical guidelines, all rats were humanely euthanized at the conclusion of the experiments using an intraperitoneal injection of sodium pentobarbital (100–200 mg/kg). This method is widely accepted and adheres to relevant animal research ethical standards, effectively reducing animal distress.

## 2.3 TTC staining

The rats were sacrificed, and the whole rat brains were collected and stored at -20˚C for 30 minutes. The brains were then cut into 2 mm thick coronal slices. The brain slices were subsequently stained with 2% TTC at 37˚C in the dark for 30 minutes, with flipping every 8 minutes to ensure uniform exposure to the TTC staining solution. Images were acquired using a digital camera (Canon EOS 800D) under consistent lighting conditions. We used ImageJ to measure the infarct area and calculated the infarct volume percentage as the infarct volume/total volume of the contralateral hemisphere × 100%.

## 2.4 Transcriptome sequencing

RNA was extracted and purified from the cerebral cortex of rats using the TRIzol reagent. The purity, concentration, and integrity of the RNA samples, as well as the presence of genomic DNA contamination, were assessed using a Nanodrop spectrophotometer, a Qubit 2.0 fluorometer, an Agilent 2100 Bioanalyzer, and electrophoresis to ensure the quality of the non-coding RNA sequencing samples. For whole transcriptome sequencing, two types of libraries were constructed. After the samples passed quality control, a starting amount of 1.5 μg of RNA was used, and the volume was adjusted to 6 μL with nuclease-free water. The Illumina Small RNA Sample Prep Kit was used to construct the miRNA library. After library preparation, high-throughput sequencing was performed on the Illumina NovaSeq platform using a single-end configuration with a read length of 1×50 bp. For the mRNA and lncRNA libraries, rRNA was removed from the samples using the Ribo-Zero™ rRNA Removal Kit (Epicentre). The purified double-stranded cDNA then underwent end repair, A-tailing, and adapter ligation, followed by fragment size selection using AMPure XP beads. Finally, the U-containing chains were degraded, and the cDNA library was obtained through PCR enrichment. Sequencing was performed on the Illumina NovaSeq platform.

## 2.5 Identification of lncRNAs

The prediction of lncRNAs involves basic screening and assessment of potential coding capacity. After basic filtering, we obtained the transcript sequence information. The basic screening consists of three parts: selecting transcripts with class_code "i", "x", "u", "o", and "e" [14]; selecting transcripts with length ≥200 bp and number of exons ≥2 [15]; and selecting transcripts with FPKM ≥0.1. We subsequently used coding potential calculators, such as the Coding Potential Calculator (CPC) [16], the Coding-Non-Coding Index (CNCI), and the Coding Potential Assessment Tool (CPAT), to filter out transcripts with coding potential (CPC score < 0.5; CNCI score < 0; CPAT score < 0.5) [17]. Furthermore, using PFAM (PFAM score < 0) protein domain analysis, we removed transcripts containing known protein domains. After prediction by four software tools, we obtained four types of lncRNAs: intergenic lncRNAs, intronic lncRNAs, sense lncRNAs, and antisense lncRNAs. RNA-seq reads were aligned to the reference genome of Rattus norvegicus using the HISAT2 tool. Following alignment, StringTie was employed for transcript assembly and to estimate the expression levels of lncRNAs.

The reference genome of Rattus norvegicus was used for sequence alignment and subsequent analysis. Newly identified genes were annotated through sequence alignment using the BLAST software against the NR, Swiss-Prot, GO, COG, and KEGG databases. Additionally, gene annotation was performed using the Ensembl database (http://asia.ensembl.org/), which facilitated the conversion of Ensembl IDs to gene symbols.

## 2.6 MiRNA identification

Based on the sequencing method outlined in Section 2.4, we successfully obtained miRNA sequencing data. Initially, we conducted rigorous quality control on the generated reads, eliminating low-quality reads and adapter sequences to yield high-quality clean reads. We then aligned these clean reads to the rat reference genome (Rattus norvegicus Rnor_5.0) to acquire genomic location information for the mapped reads. Subsequently, we employed alignment software, such as Bowtie, to compare the mapped reads with multiple databases (e.g., Silva, GtRNAdb, Rfam, Repbase), filtering out non-coding RNAs, including rRNA, tRNA, snRNA, snoRNA, and repetitive sequences, to obtain unannotated reads.

During the miRNA identification process, we aligned the mapped reads with known mature miRNA sequences from the miRBase (v22) database, permitting a maximum of one mismatch to identify known miRNAs. For sequences that could not be identified, we utilized the miRDeep2 software to predict novel miRNAs. miRDeep2 analyzes the distribution of mapped reads across the genome (including the mature strand, star strand, and hairpin structure) and the energy information of precursor structures, applying a Bayesian model for scoring to identify potential miRNA precursor sequences and make predictions.

## 2.7 MRNA identification

After alignment to the Rattus_norvegicus reference genome using HISAT2, transcript assembly was performed using StringTie. mRNAs were identified using two approaches: To identify mRNAs, two approaches can be utilized. The first involved the use of a Perl script to search for mRNAs within a range of 100 kilobases (kb) upstream or downstream of lncRNAs [18]. The second approach employs the Pearson correlation coefficient to assess the relationship between lncRNA and mRNA expression across samples, identifying mRNAs with an absolute correlation coefficient exceeding 0.9 and a P value less than 0.01 [19].

## 2.8 Identification of differentially expressed lncRNAs, miRNAs and mRNAs

Differential expression analysis of lncRNAs, miRNAs, and mRNAs was performed using the DESeq R package (version 1.10.1). The significance of the P value obtained from the original hypothesis test reflects the probability of observing no difference in expression. To control for multiple testing, the Benjamini–Hochberg correction method was applied to adjust the significant P values obtained from the original hypothesis test. The false discovery rate (FDR) was subsequently utilized as the criterion for screening differentially expressed lncRNAs, miRNAs, and mRNAs. Specifically, for differentially expressed lncRNAs and mRNAs, fold change (FC) $\geq 2$, an FDR < 0.01 [20], and P value < 0.05 were used as screening criteria. For differentially expressed miRNAs, $|\log2FC|$ [21] $\geq 2$, FDR < 0.01, and P value < 0.05 were employed as the screening criteria.

## 2.9 Prediction of the subcellular localization of lncRNAs

We employed lncLocator software (http://www.csbio.sjtu.edu.cn/bioinf/lncLocator/) to predict the subcellular localization of these lncRNAs. The LncLocator algorithm assigns prediction scores for potential subcellular localizations of each lncRNA, including the cytoplasm, nucleus, ribosome, cytosol, and exosome. The location with the highest score is designated the predicted subcellular localization.

## 2.10 Functional enrichment analysis

GO and KEGG analyses were performed using the Database for Annotation, Visualization, and Integrated Discovery (DAVID, https://david.ncifcrf.gov/).). and GO analysis was employed to investigate the primary functions of the DEmRNAs. GO terms were further divided into biological process (BP), cellular compone (CC), and molecular function (MF) categories. KEGG pathway enrichment analysis plays a crucial role in elucidating biological mechanisms and functions. Functional enrichment analysis was performed at a significance level of P < 0.05. Through DAVID analysis, we identified the top 10 significantly enriched GO terms and KEGG pathways.

## 2.11 CeRNA network analysis

The miRDB database (http://www.mirdb.org/) was used to predict miRNAs corresponding to lncRNAs with a minimum target score of 60 and conservation scores > 0.5. TargetScan (http://www.targetscan.org/) and miRanda (http://www.microrna.org/) were used to screen for mRNAs that are complementary to the miRNAs, with TargetScan context++ scores $\leq$ -0.4 and miRanda scores > 140 (energy < -20 kcal/mol). Only targets predicted by both algorithms were retained. The ceRNA network of lncRNAs/miRNAs/mRNAs was visualized using Cytoscape (version 3.9.1).

## 2.12 Construction of the PPI network

The PPI networks were constructed using the STRING (https://string-db) database, and the confidence scores were set to $\geq$0.4. Cytoscape software was used to visualize the results. According to cytoNCA, a Cytoscape plugin, the top 20 DEmRNAs were selected as the hub genes for further analysis and research.

## 2.13 Validation of the RNA-seq results using quantitative PCR (qPCR)

Total RNA was extracted from the infarcted cortex and hippocampal tissues using TRIzol reagent (Invitrogen, Carlsbad, CA, USA) according to the manufacturer's instructions, and the

integrity of the RNA was confirmed by 1% agarose gel electrophoresis. RNA purity was assessed using a NanoDrop 2000 spectrophotometer (Thermo Fisher Scientific). Complementary DNA (cDNA) was synthesized from lncRNA using M-MLV reverse transcriptase (Promega) with 5 μl of RNA in a 20 μl reaction volume. Nested PCR was employed for exponential amplification of the target fragments. In the first round of PCR, 1 μl cDNA was used as the template, with an annealing temperature of 50˚C for 40 cycles. The first-round PCR product was then diluted 100-fold, and 1 μl of this diluted product was used as the template for the second round of PCR, with an annealing temperature of 55˚C for 40 cycles.Reverse transcription of mRNA and miRNA was performed using the PrimeScript™ RT reagent kit (TaKaRa Bio, Kusatsu, Japan) and the miRNA Reverse Transcription Kit (TaKaRa Bio), respectively. qPCR was conducted using TB Green dye (TaKaRa) on a Roche LC96 real-time system under the following conditions: 95˚C for 120 seconds (preincubation, 1 cycle), 95˚C for 15 seconds, 60˚C for 30 seconds, and 72˚C for 10 seconds (amplification, 40 cycles). ACTB was used as an internal standard for lncRNAs and mRNAs, and U6 was used as an internal standard for miRNAs. The results were analyzed using the 2−ΔΔCt method. The primer sequences for the lncRNAs, miRNAs, and mRNAs are listed in S1 Table.

### 2.14 Statistical analysis

For all the lncRNA, miRNA, and mRNA sequencing data, logarithmic transformation (base 2) was applied to approximate a normal distribution for the normality test (Shapiro–Wilk). Statistical analyses were performed using R v3.5.3, and the Benjamini–Hochberg method was used to correct the P values in the differential screening. A significance threshold of $P < 0.05$ was applied for statistical significance.

## 3 Results

### 3.1 Successful establishment and assessment of the MCAO model

After 2 h of MCAO followed by 24 h of reperfusion (MCAO 2 h/R 24 h), we evaluated the results using neurobehavioral scoring and morphological assessment. In the neurobehavioral evaluations, the rats in the MCAO group exhibited an inability to extend their forelimbs and a lack of desire to grasp or land after cerebral infarction. In contrast, the sham-operated rats were able to naturally extend their forelimbs and showed a strong desire to grasp the ground (Fig 2A and 2B). The Longa score further revealed more obvious neurological deficits in the MCAO group than in the sham-operated group (Fig 2C). Morphologically, TTC staining of whole-brain sections from the MCAO group revealed pale infarct areas and severe edema, whereas the brain tissue from the control group was ruddy in color without obvious infarct lesions (Fig 2D and 2E). These results indicated that the experimental models had been successfully established and were suitable for subsequent sequencing analysis.

### 3.2 Identification and characterization of lncRNAs in the cortex

Total RNA was extracted from infarcted cortical tissue, and its purity was assessed, with OD260/280 $\geq$ 1.8 and OD260/230 $\geq$ 1.0. High-throughput sequencing was conducted on five cortical samples from the MCAO group (T01—T05) and five samples from the sham group (T06—T10) (Fig 3). After low-quality sequences were removed, the number of clean read data for both the experimental and control samples exceeded 65 million reads. The average GC content of all the samples was approximately 47%, and the Q30 ratio exceeded 93.75%. These clean reads were aligned to the rat reference genome, and the mapping rate of the lncRNA reads exceeded 95% (S2 Table), indicating the reliability of the RNA sequencing data. Using

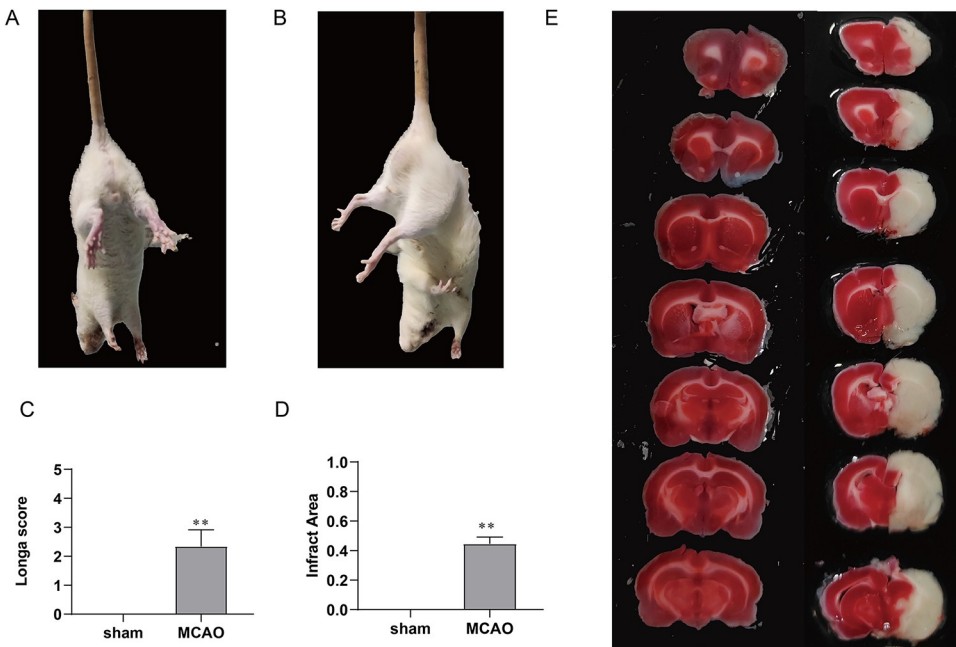

**Fig 2. Establishment and evaluation of the MCAO model of brain injury after ischemia–reperfusion.** (A) Behavioral observation of sham-operated rats. (B) Behavioral observation of MCAO rats. (C) The Longa score assessment. (D) The quantification Quantification of the infarct area. (E) TTC-stained brain sections. The left image shows the sham group, and the right image illustrates the MCAO group, with the pale ischemic area visible in the right brain. The data are presented as the means ± SEMs and were analyzed using an independent sample t-test (**$P < 0.01$ vs. the sham group, n = 3).

predictions from four potential coding ability software tools, a total of 31,183 lncRNA transcripts were identified, and their identifiers are listed in S3 Table. Based on their genomic locations, the lncRNAs were further categorized into different classes, including 19,848 (63.7%) sense lncRNAs, 2,644 (8.5%) antisense lncRNAs, 6,432 (20.6%) intronic lncRNAs, and 2,259 (7.2%) intergenic lncRNAs (lincRNAs) (S1A Fig).

To screen for reliable DELs, consistency and dispersion checks were performed on the sequencing data of each sample. The box plot revealed that the expression levels of the samples were relatively concentrated, with no obvious outliers, indicating the reproducibility of the sequencing data (Fig 4A). By applying filters of FC ≥2 and FDR < 0.01, a heatmap and volcano plot of DELs were generated (Fig 4B and 4C). A total of 551 DELs were identified between the sham and MCAO groups, including 416 upregulated and 135 downregulated lncRNAs (S4 Table). The counts, FPKM values, and FDR values of each DEL among the ten sequencing samples are detailed in S5 Table.

To prioritize the most critical DELs for further research, we first performed an analysis that focused on the intersection of the top 200 expression levels and the top 200 differences in significance among 551 DELs. This analysis identified 62 DELs (S6 Table). From this subset, we then selected the top 20 lncRNAs based on expression levels and another top 20 based on significant differences, thus identifying key lncRNAs. This process resulted in 33 unique lncRNAs. Using the lncRNA sequencing data, we subsequently employed lncLocator to determine the subcellular localization of these lncRNAs. The prediction results indicated that among the 33 lncRNAs, 25 were located in the cytoplasm, and 8 were located in the nucleus (S7 Table). Finally, we designed primers targeting the predicted cytoplasmic lncRNAs to verify whether their expression trends were consistent with the sequencing results (Fig 5A). Using

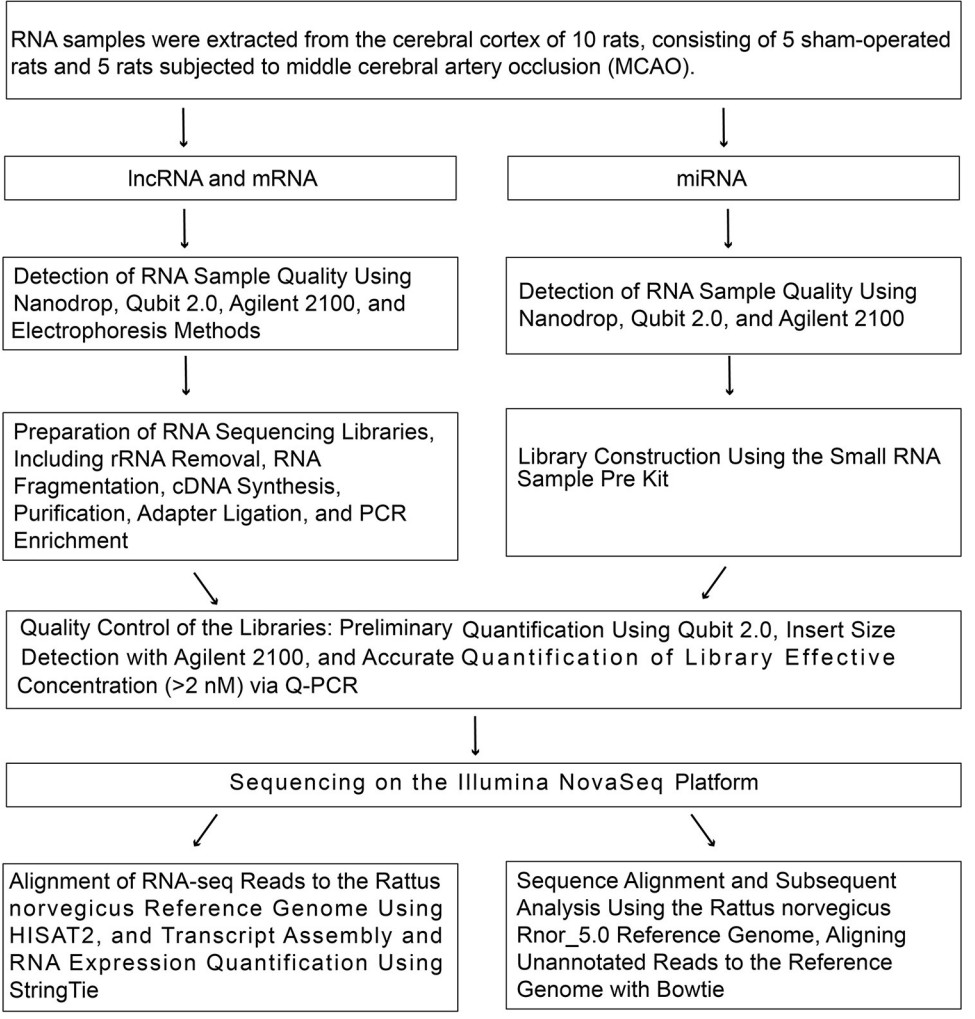

**Fig 3. High-throughput sequencing of lncRNAs, miRNAs, and mRNAs.**

RT–PCR, we successfully validated the upregulation of 7 selected lncRNAs (Fig 5C). Additionally, we randomly selected 4 lncRNAs whose expression did not significantly change (Fig 5B) and 6 downregulated lncRNAs (Fig 5D) for RT–qPCR validation. The results demonstrated that the expression patterns of these genes were consistent with those obtained from RNA-Seq.

### 3.3 Identification of differentially expressed miRNAs and mRNAs

For miRNAs, each sample contained no fewer than 12.04 million clean reads. Among these genes, 59.59% (MCAO group) and 69.21% (sham group) were successfully mapped to the rat reference genome (S8 Table). Using the miRBase (v22) database and miRDeep2 software, we identified 2,145 miRNAs, 19.91% of which were known miRNAs. By applying thresholds of adjusted $p < 0.05$ and $|log2FC| > 1$, we identified 651 DEmiRNAs between the MCAO and sham groups, with 464 upregulated and 187 downregulated miRNAs (S9 Table) (Fig 6A and 6B). Additionally, using stricter criteria of $|log2FC| > 2$ and FDR $< 0.01$, we identified 2,608 DE mRNAs, including 1,133 upregulated and 1,475 downregulated mRNAs (S10 Table) (Fig 6C and 6D).

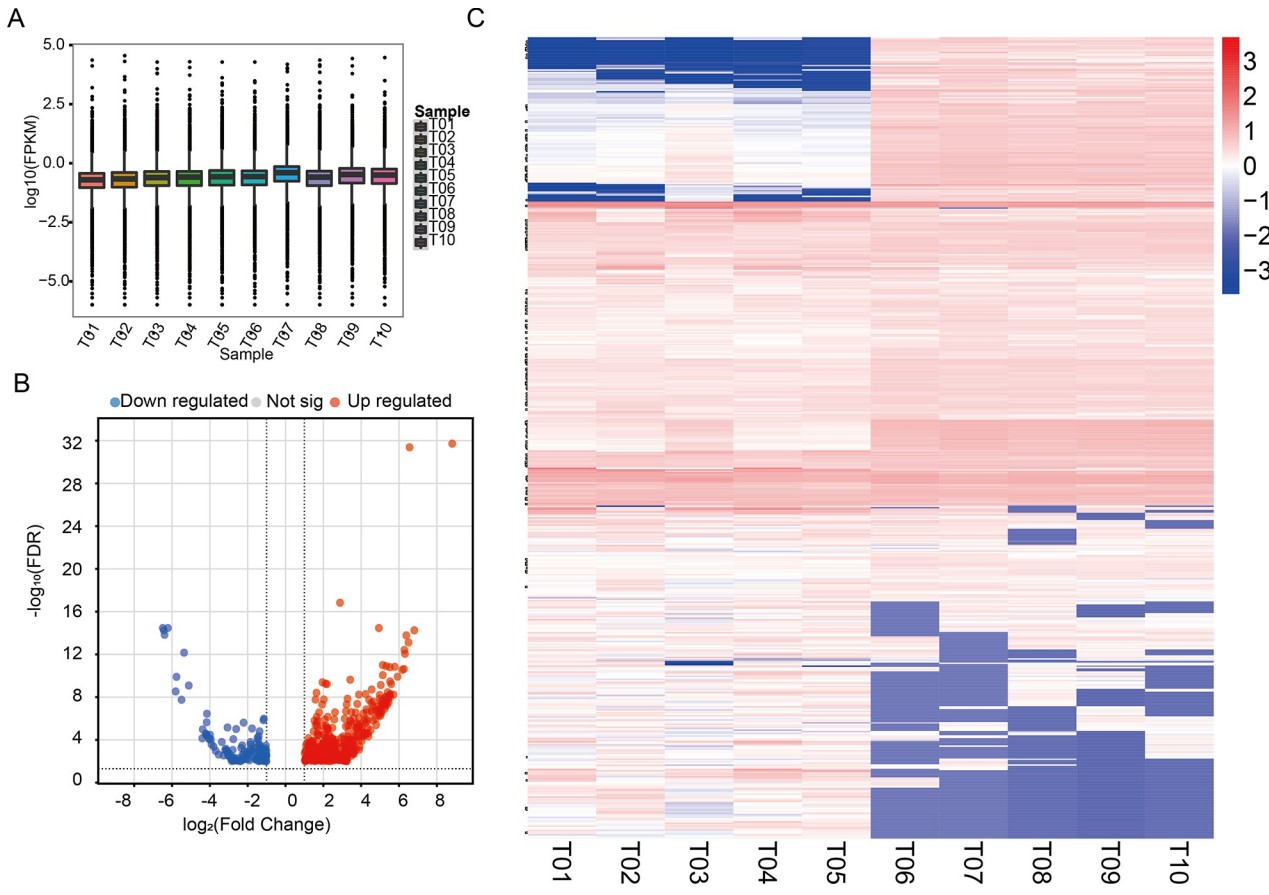

**Fig 4. Expression analysis of lncRNAs between the MCAO and sham groups.** (A) The box plot illustrates the expression abundance of lncRNAs in each sample. The x-axis represents the samples, and the y-axis represents the logarithm value of normalized sample expression using the spliced reads per billion mapping (RPB) algorithm. (B) The volcano plot shows DELs between the two groups. The blue and red dots represent downregulated and upregulated DELs, respectively, in the MCAO group compared with the sham group (FDR < 0.01). (C) A clustering heatmap illustrates distinguishable expression profiles of lncRNAs. The colors from blue to red indicate an increase in relative lncRNA expression. T01–T05: MCAO group; T06–T10: sham group.

### 3.4 Construction of the ceRNA network in stroke

By intersecting the top 20 lncRNAs on the basis of expression levels with the top 20 lncRNAs on the basis of differential significance, we identified 25 lncRNAs localized in the cytoplasm. These lncRNAs were predicted to interact with 493 miRNAs using the miRDB [22] database. We analyzed the differential expression of 493 predicted miRNAs in the MCAO model. From the 651 miRNAs identified using high-throughput sequencing, we identified 59 that were differentially expressed. These 59 miRNAs interacted with 21 lncRNAs, excluding 4 lncRNAs (MSTRG.128663.1, MSTRG.60561.1, MSTRG.97892.39, and MSTRG.113905.1) whose predicted miRNAs did not significantly differ from those in the sham group. According to the ceRNA theory, lncRNAs and miRNAs are thought to exhibit inverse expression patterns, particularly in lncRNAs that harbor miRNA response elements (MREs). This theory proposes that lncRNAs act as molecular sponges by competitively binding miRNAs through shared MREs, thereby reducing the availability of miRNAs to bind their mRNA targets. As a result, this competition can indirectly regulate mRNA stability and translation. In the aforementioned lncRNA–miRNA network, after 22 miRNAs whose expression trends were the same as those of the lncRNAs were removed, we identified 50 lncRNA–miRNA pairs, including 20

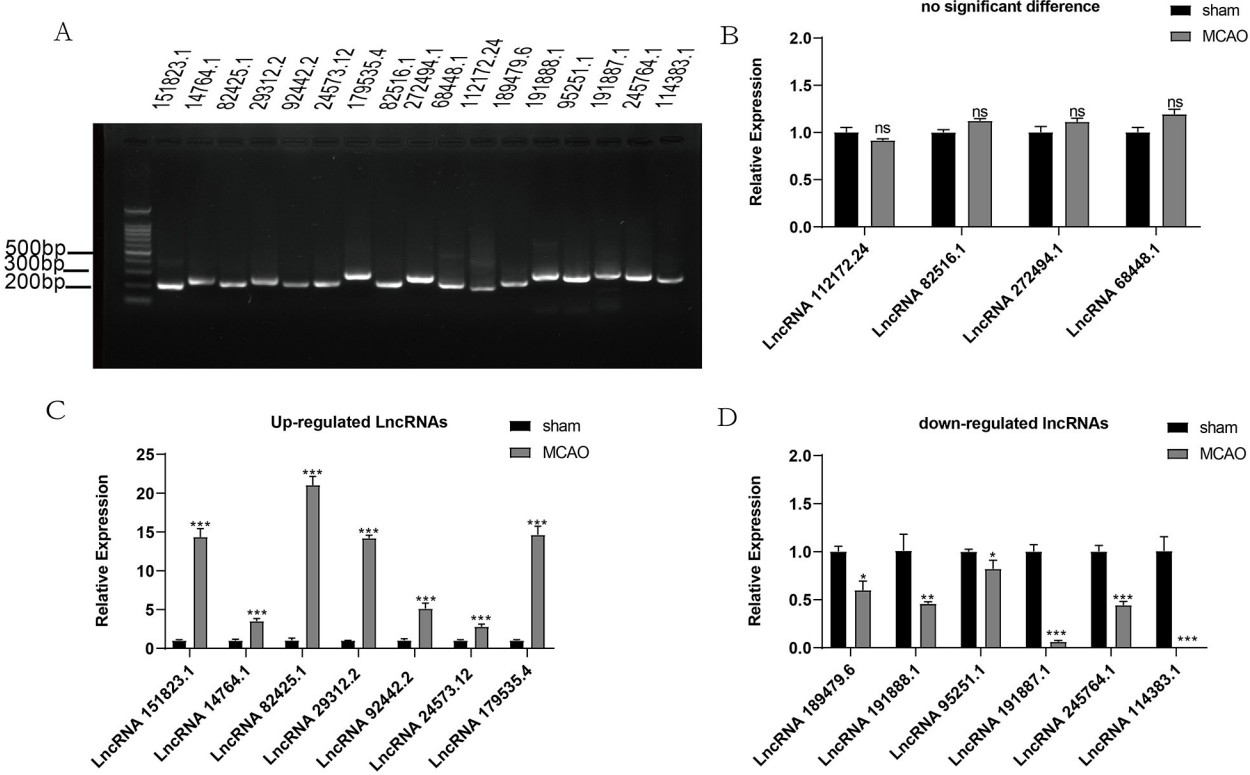

**Fig 5. Identification of representative lncRNAs and validation of their expression.** (A) Agarose gel electrophoresis was used to determine the sizes of the lncRNA PCR products. The first well of the agarose gel contained the marker, while the subsequent wells contained the lncRNA samples. (B) qRT–PCR was performed to confirm the expression of lncRNAs, the results of which were not significantly different. (C) qRT–PCR was conducted to validate the upregulated expression of lncRNAs. (D) qRT–PCR was performed to confirm the downregulated expression of lncRNAs. The data are presented as the mean ± SEM (n = 3). ns (no significant difference), *$P < 0.05$, **$P < 0.01$, ***$P < 0.001$ vs. the sham group, as determined by an independent sample t-test.

lncRNAs and 37 miRNAs (S11 Table). Using Cytoscape software, we constructed a network comprising 50 lncRNA–miRNA pairs (Fig 7).

We utilized miRanda [23] and TargetScan [24] to predict the target mRNAs of 37 differentially expressed miRNAs. To confirm the differentially expressed target mRNAs, we intersected the predicted target mRNAs with the 2608 differentially expressed mRNAs (DEmRNAs) identified through high-throughput sequencing in the MCAO model. This analysis resulted in the identification of 634 differentially expressed target mRNAs. After excluding 443 mRNAs whose expression trends were consistent with the miRNA expression trends, we retained 16 miRNAs (excluding 11 miRNAs whose predicted mRNAs did not show differential expression compared with the sham group and excluding 10 miRNAs whose expression trends were consistent with those of the mRNAs) and 191 mRNAs (S12 Table). Finally, we used Cytoscape software to construct a network comprising 232 miRNA–mRNA interactions (Fig 8).

We successfully constructed a lncRNA-based ceRNA regulatory network using Cytoscape. This network comprised 12 lncRNAs, 16 miRNAs, and 191 mRNAs (S13 Table) (Fig 9).

## 3.5 Protein–protein interaction network assessment and functional enrichment analysis of DEmRNAs

We constructed a PPI network containing 167 nodes and 196 edges. These 167 nodes represent genes encoding known proteins. By performing CytoNCA analysis, we identified the top

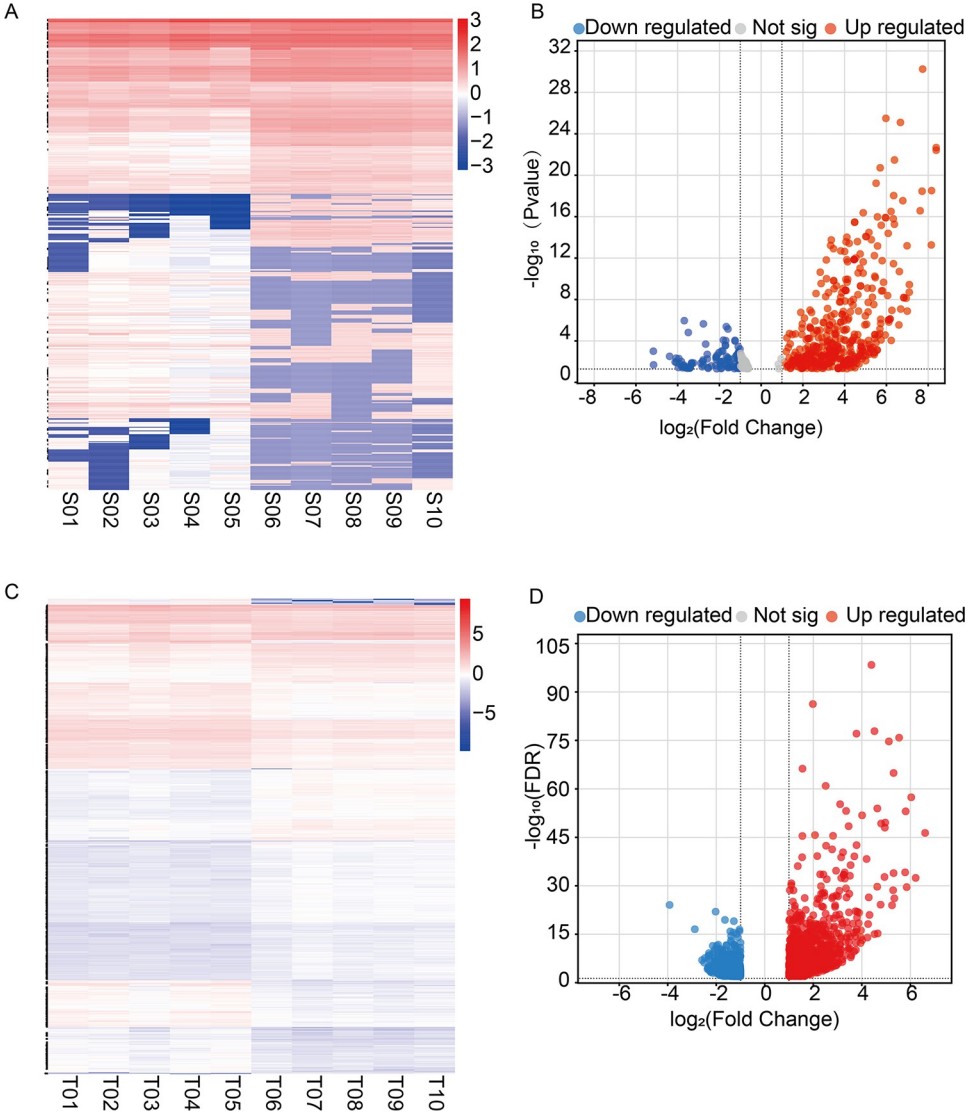

**Fig 6. Identification of differentially expressed miRNAs and mRNAs.** (A) and (C):The heatmaps show differentially expressed miRNAs and mRNAs. S01–S05 and T01–T05 are samples from the MCAO group, whereas S06–S10 and T06–T10 are samples from the sham group. The color scale from blue to red indicates increasing expression levels. (B) and (D): The volcano plots compare the MCAO group to the sham group. The blue dots indicate downregulated miRNAs/mRNAs, the red dots indicate upregulated miRNAs/mRNAs, and the gray dots indicate no significant differential expression.

20 genes with the highest scores as hub genes (S14 Table). The top 20 hub genes are highlighted in red in Fig 10. The BP terms identified through GO analysis of the DEGs were significantly enriched in angiogenesis, the cytokine response, and cytokine-mediated signaling pathways (S15 Table) (Fig 11A). In the CC analysis, we observed significant enrichment in the basement membrane, extracellular matrix, and collagen-containing extracellular matrix (S16 Table) (Fig 11B). Integrin binding, cytokine receptor activity, and extracellular matrix structural constituents were markedly prominent within the MF category (S17 Table) (Fig 11C). KEGG pathway analysis revealed that the DEGs were associated primarily with the PI3K-Akt signaling pathway, ECM-receptor interaction, and NF-kappa B signaling pathway (S18 Table) (Fig 11D). The enrichment of ECM-receptor interactions, which was consistent with the GO

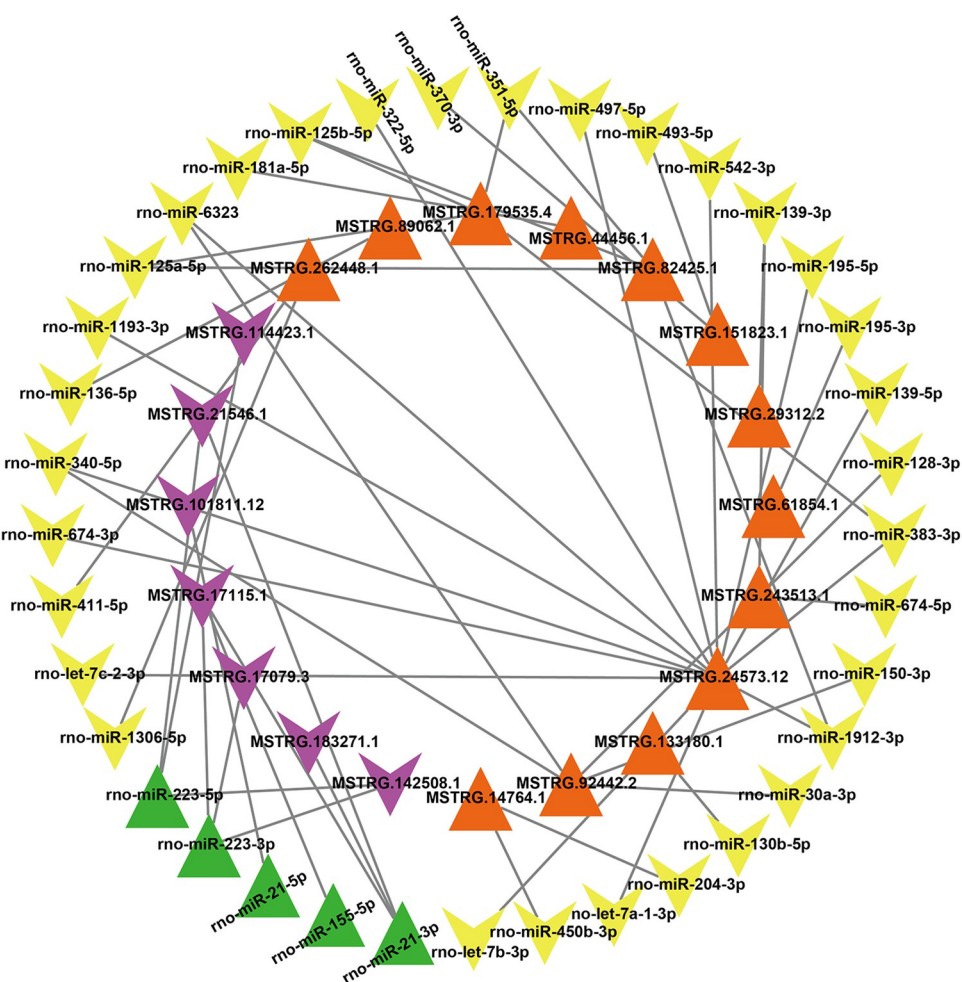

**Fig 7. The lncRNA–miRNA network.** The green nodes represent upregulated miRNAs, whereas the yellow nodes represent downregulated miRNAs. Similarly, orange nodes represent upregulated lncRNAs, and purple nodes represent downregulated lncRNAs compared with those in the sham group (p < 0.05). Each edge in the network indicates an interaction between two nodes.

analysis results, highlighted the significant role of interactions between the extracellular matrix and cell membrane receptors in the cerebral vascular and nervous systems, potentially involving neural protection and repair [25]. The NF–kappa B signaling pathway plays a critical role in inflammation and immune regulation, which may be relevant to the inflammatory response and damage repair in stroke [26].

## 3.6 RT–qPCR Validation of DE miRNAs and DE mRNAs

Among the top 20 hub genes identified, 11 have been reported to be associated with IS: Furin, Tacc3, Lgals3, Ptgs2, Il6r, Bag3, Serpine1, Flna, Jak3, Myh9, and Mcm2. Based on these 11 hub genes, we constructed the core subnetwork of the ceRNA network, comprising 4 lncRNAs, 4 miRNAs, and 11 mRNAs. We performed RT–qPCR validation of the DEmRNAs and DEmRNAs. In the miRNA assays, all the miRNAs were downregulated, with miR-370-3p and miR-139-3p exhibiting significant reductions in expression exceeding 50%. Conversely, the mRNA assays revealed the upregulation of all mRNAs, with Bag3, Serpine1, and Flna demonstrating significantly elevated expression levels, each exceeding a threefold increase. Notably, miR-370-

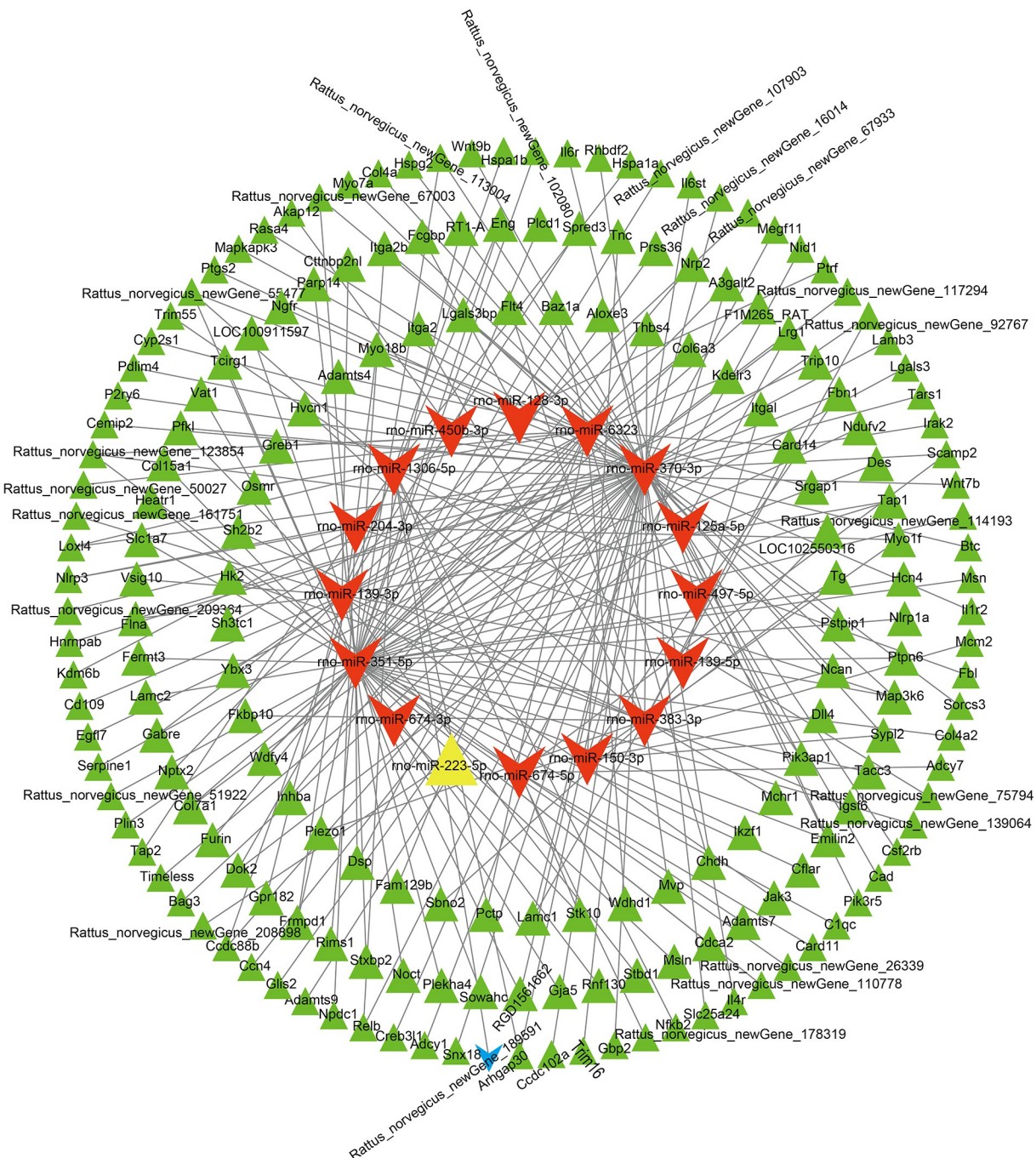

**Fig 8. The miRNA–mRNA network.** The yellow nodes represent upregulated miRNAs, whereas the red nodes represent downregulated miRNAs. Similarly, green nodes represent upregulated mRNAs, and blue nodes represent downregulated mRNAs, all in comparison to the sham group (p < 0.05). Each edge in the network indicates an interaction between two nodes.

3p negatively regulates all three mRNAs. Finally, for the 4 lncRNAs, MSTRG.82425.1, MSTRG.151823.1, and MSTRG.29312.2 were upregulated by more than tenfold. The expression patterns of the lncRNAs, miRNAs, and mRNAs aligned with the ceRNA network predictions. Fig 12 further confirms the reliability and validity of the sequencing data.

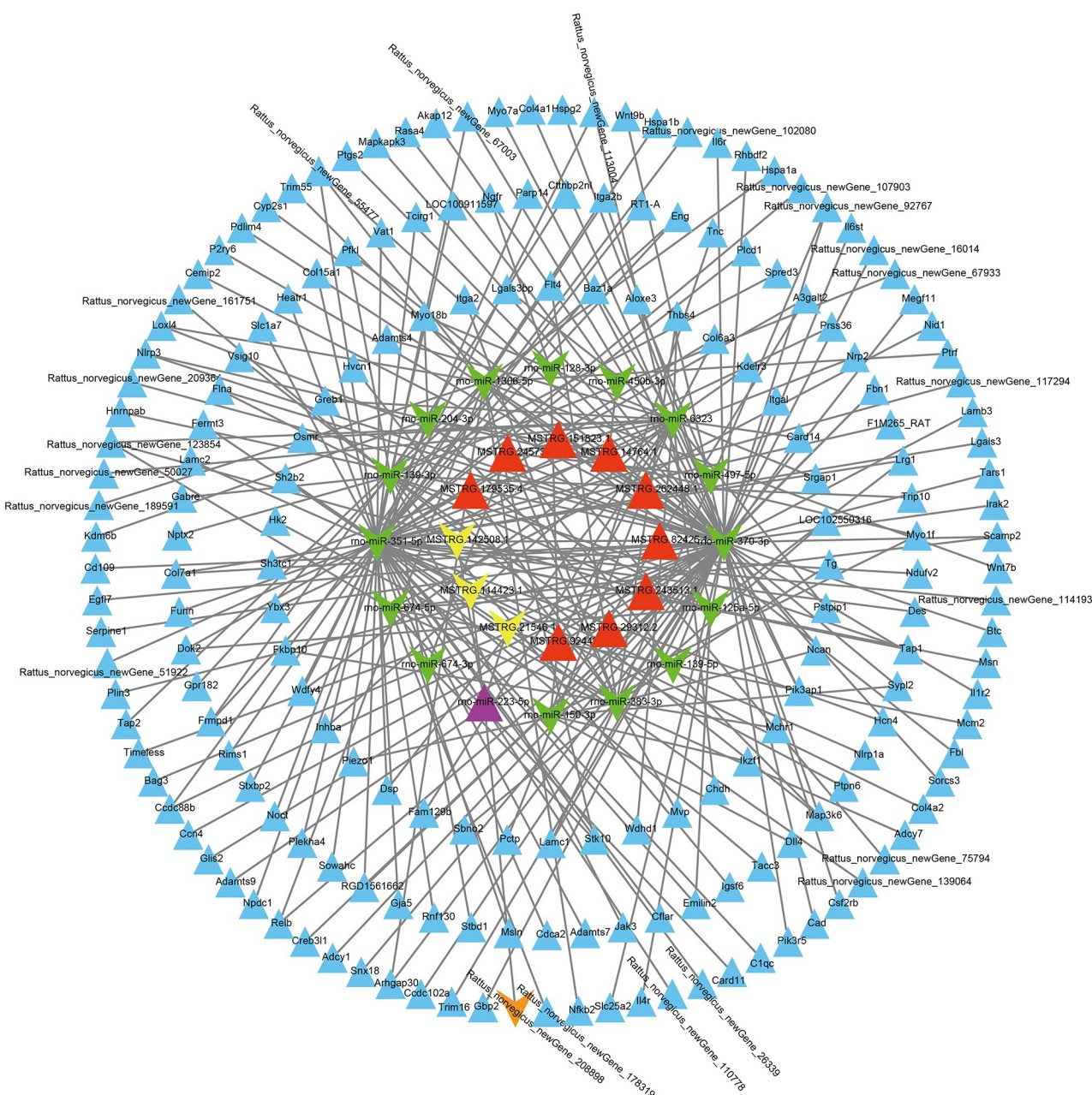

**Fig 9. In the competitive endogenous RNA (ceRNA) network involving lncRNAs, miRNAs, and mRNAs, node colors indicate the expression status of different RNA types.** In the network visualization, red nodes represent upregulated lncRNAs, yellow nodes represent downregulated lncRNAs, purple nodes represent upregulated miRNAs, green nodes represent downregulated miRNAs, blue nodes represent upregulated mRNAs, and orange nodes represent downregulated mRNAs compared with the sham group (p < 0.05). Each edge in the network denotes an interaction between two nodes.

### 3.7 Biological pathways of key LncRNAs and their regulatory roles in ischemic stroke

Following the validation of 4 miRNAs and 11 mRNAs, we identified 4 lncRNAs and subsequently constructed a regulatory network involved in poststroke pathophysiological processes. This regulatory network may be implicated in various biological events involved in ischemic

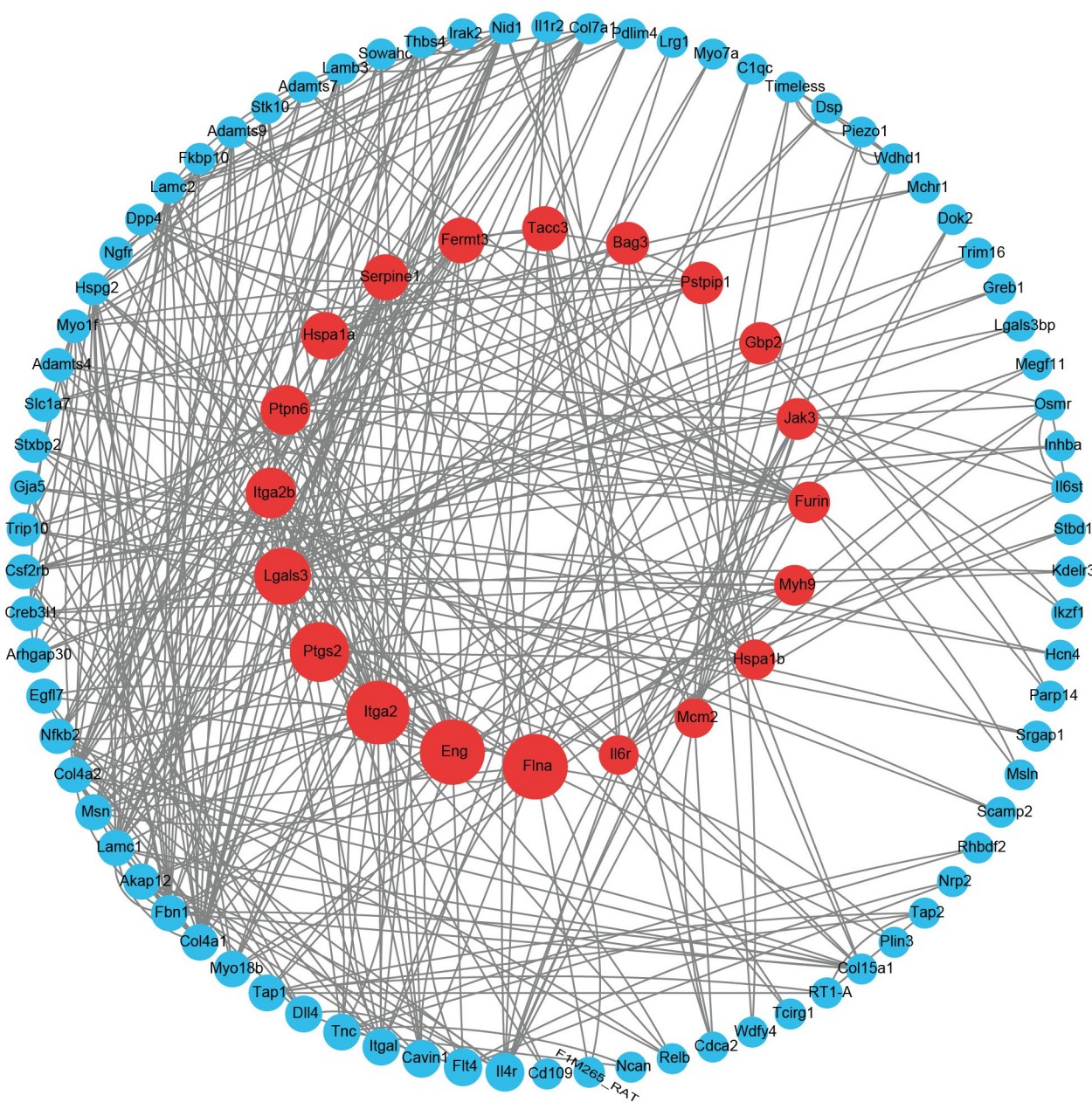

**Fig 10. The PPI network and screening of the top 20 hub genes.** The genes marked in red represent the top 20 hub genes.

stroke, including angiogenesis, axonal regeneration, poststroke inflammation, microglial activation, BBB disruption, apoptosis, autophagy, ferroptosis, and thrombocytopenia. Our study demonstrated that MSTRG.151823.1 modulates the expression of multiple apoptosis-related genes, including Bag3, Serpine1, and Il6r, following stroke. We determined that Bag3 and Serpine1 are potentially upregulated as molecular targets of miR-370-3p, thereby suppressing cell apoptosis and facilitating cell recovery. Conversely, MSTRG.151823.1 promotes cell apoptosis by downregulating miR-370-3p, resulting in the upregulation of Il6r. With respect to angiogenesis, MSTRG.82425.1, MSTRG.151823.1, MSTRG.29312.2, and MSTRG.243513.1 participate in the regulation of angiogenesis. Increased levels of MSTRG.82425.1 could downregulate

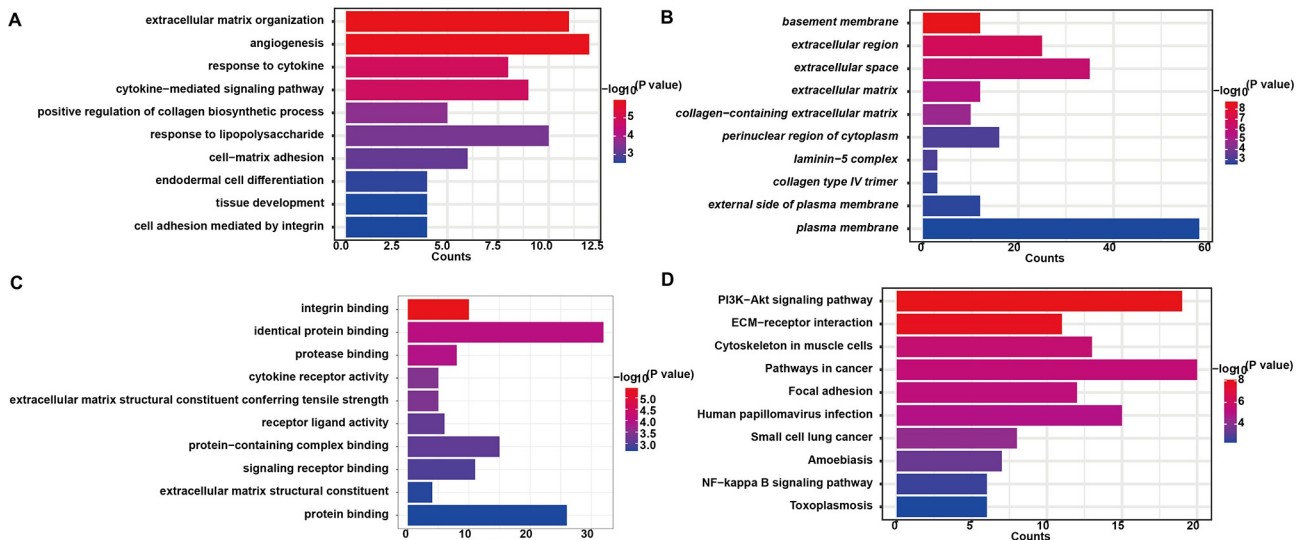

**Fig 11. GO and KEGG pathway analyses of the DE-mRNAs.** (A-C) The top 10 enriched biological processes (BP), cellular components (CC), and molecular functions (MF) of the common differentially expressed messenger RNAs (DE-mRNAs). (D) The KEGG pathway analysis of the common DE-mRNAs. The color transition from blue to red indicates decreasing *P* values, indicating increasingly significant differences. The bar lengths, from short to long, represent an increasing number of enriched genes.

miR-351-5p, leading to increased Furin expression and the subsequent promotion of vascular remodeling. Furthermore, MSTRG.151823.1 downregulates miR-370-3p, whereas MSTRG.29312.2 and MSTRG.243513.1 may function as molecular sponges for miR-139-3p, exerting a synergistic effect. The downregulation of miR-370-3p and miR-139-3p subsequently enhances the expression of Flna following stroke, thereby facilitating angiogenesis. Conversely, MSTRG.151823.1 suppresses angiogenesis by downregulating miR-370-3p, which targets Ptgs2 and Serpine1. Our findings indicate that the upregulation of Ptgs2, Lgals3, Il6r, and Jak3 further enhances the inflammatory response following stroke (Fig 13).

## 4 Discussion

Ischemic injury to brain tissue leads to neuronal cell death and injury, triggering a cascade of pathophysiological processes, including inflammatory responses, BBB disruption, oxidative stress, pyroptosis, and apoptosis [27]. During the pathological process of ischemic stroke, the activation of microglia serves as an initial response, triggering a sequence of biological events. Microglia release inflammatory factors, inducing endoplasmic reticulum stress, which affects protein folding and leads to cellular dysfunction [28, 29]. This stress response further damages the BBB, increasing its permeability and leading to cell death and ferroptosis, which in turn exacerbates neuronal loss [27]. Cell death primarily occurs through two pathways: apoptosis, a programmed cell death process, and necroptosis, triggered by an intracellular calcium imbalance, which may intensify local inflammatory responses. Moreover, autophagy, a protective cellular response that clears damaged organelles and inhibits apoptosis, may become harmful if excessively activated [28]. During the recovery phase of stroke, angiogenesis plays a critical role in promoting blood supply and tissue repair in damaged areas. These processes not only are interconnected but also significantly influence the progression of stroke and patient recovery [30, 31].

Extensive lncRNA-centered research has shown that nuclear lncRNAs control the epigenetic states of specific genes [32], participate in transcriptional regulation [33], contribute to

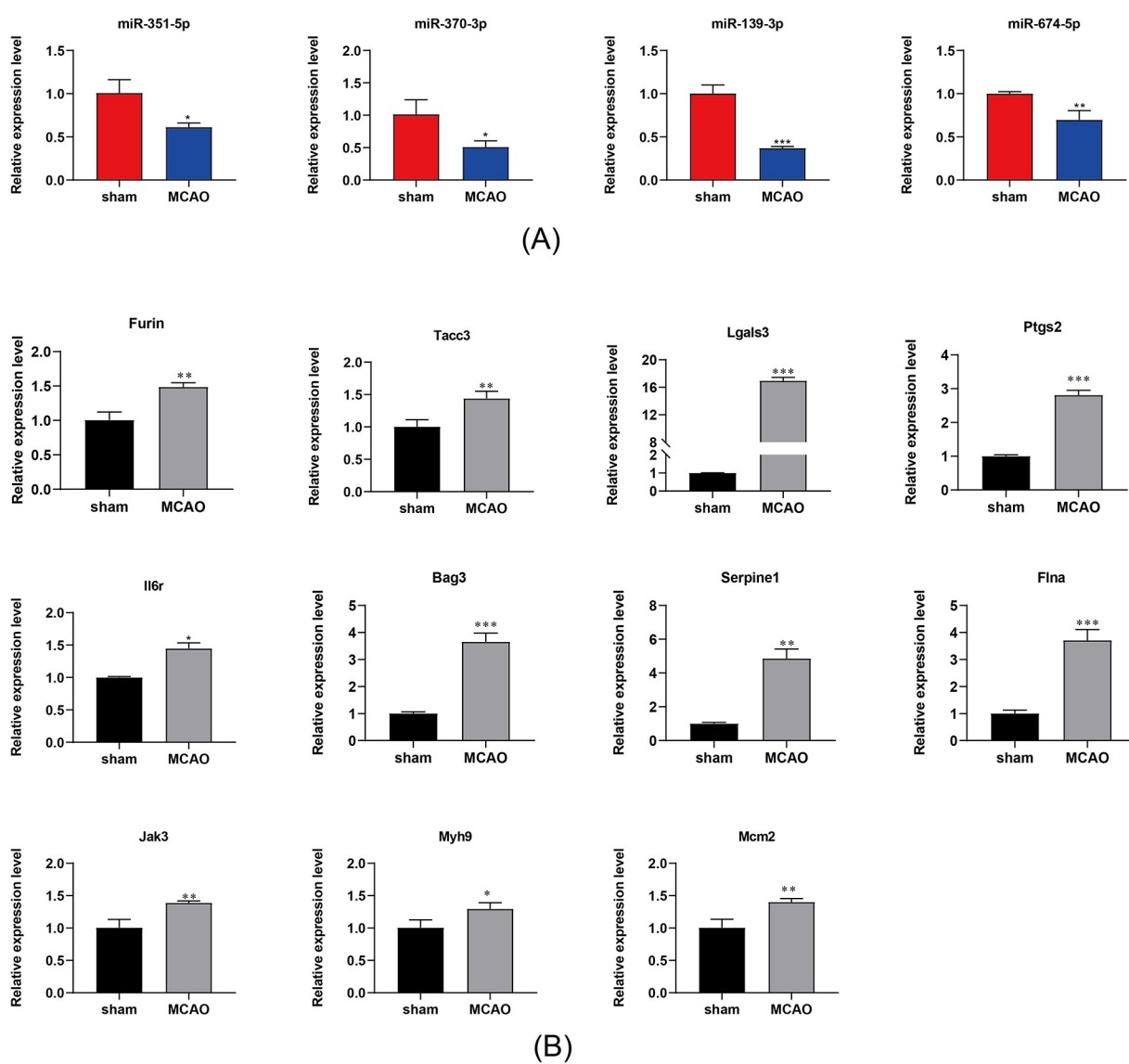

**Fig 12. RT–qPCR Validation of DE miRNAs and DE mRNAs.** (A) qPCR validation of the miRNA sequencing data. (B) qPCR validation of the mRNA sequencing data. The data are presented as the means±SEM (n = 3). * $P < 0.05$, ** $P < 0.01$ and *** $P < 0.001$; two-tailed paired t-test.

alternative splicing, and form subnuclear compartments [34, 35]. Notably, although nuclear lncRNAs are more abundant than cytoplasmic lncRNAs, they are not as stable as cytoplasmic lncRNAs [36, 37]. The cell nucleus serves as the site of biogenesis and processing for most lncRNAs, whereas the cytoplasm serves as the final residence and functional site for certain lncRNAs [35]. In the cytoplasm, lncRNAs function to mediate signal transduction pathways, translational programs, and post-transcriptional control of gene expression. For example, lncRNAs can sequester miRNAs [38] and proteins [39] to regulate their activity and levels, influence protein posttranslational modifications [40], or mediate mRNA translation and stability [41–43]. Growing evidence suggests that lncRNAs function as ceRNAs in various human diseases, including cancer, metabolic disorders, and cardiovascular diseases. For example, studies have highlighted the critical role of lncRNA-MIAT in pathological angiogenesis, in which it acts as a ceRNA, forming a feedback loop with vascular endothelial growth factor

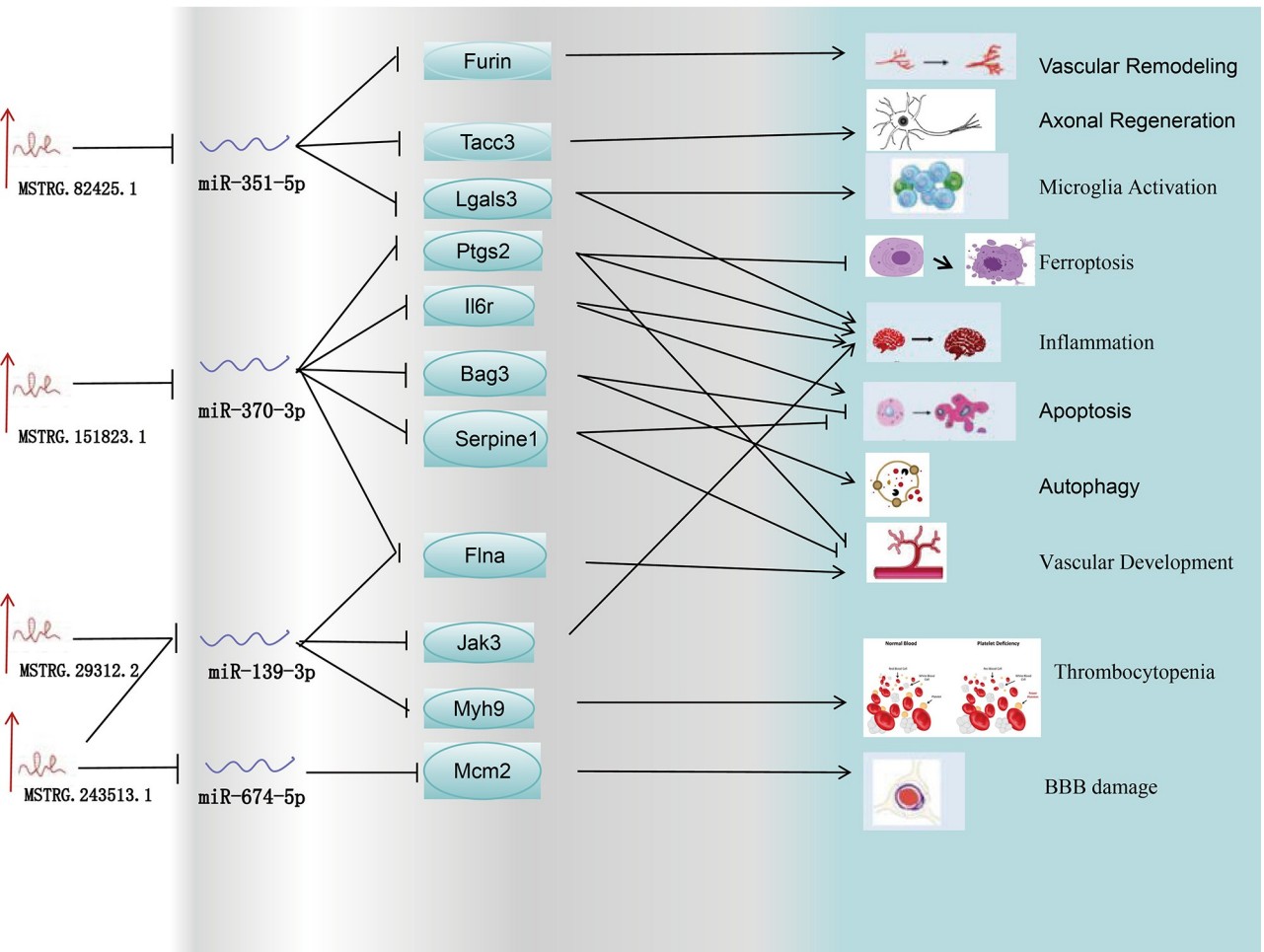

**Fig 13. The key candidate lncRNAs in ischemia–reperfusion injury and their inferred biological pathways.** The white columns list key DEGs, whereas the light gray and dark gray columns show miRNA and mRNA targets that play crucial roles in neural injury. The blue columns list the corresponding biological mechanisms confirmed in brain injury following ischemia–reperfusion.

(VEGF) and miR-150-5p and thereby modulating endothelial cell function [44]. Research has revealed that lncRNA–miRNA-mRNA interactions are involved in the pathophysiology of ischemic stroke and play important roles in neuroprotection and postischemic recovery. For example, the lncRNA FTX acts as a sponge for miR-342-3p to regulate SPI1 expression levels, enhancing angiogenesis in stroke [45]. The lncRNA NEAT1 regulates NLRP3 inflammasome activation in microglia by forming a NEAT1/miR-10b-5p/BCL6/NLRP3 axis, thereby mitigating the harmful outcomes of ischemic stroke-induced inflammation [46]. Thus, these lncRNAs derived from the ceRNA network play crucial roles in ischemic stroke.

In the MCAO model, rats were subjected to occlusion of the middle cerebral artery for 2 hours, followed by 24 hours of reperfusion. Behavioral assessments revealed significant neurological deficits, and TTC staining revealed distinct pale ischemic infarct areas. These findings suggest that this model accurately represents severe injury, which is consistent with the acute phase of stroke progression. Cortical tissues from both MCAO and sham rats were collected for deep sequencing, which yielded more than 65 million clean reads, with more than 97% mapping successfully to the rat reference genome. The uniform distribution of lncRNAs across chromosomes confirmed the consistency and reliability of the sequencing data, reflecting high

sequencing quality. Differential expression analysis using stringent criteria (FC≥2 and FDR<0.01) revealed that 416 lncRNAs were upregulated and 135 were downregulated compared with those in the sham group. We conducted a cross-analysis of the top 20 lncRNAs on the basis of expression levels and the top 20 lncRNAs on the basis of differential expression significance to predict the subcellular localization of these intersecting lncRNAs, revealing that 25 are localized in the cytoplasm. We subsequently selected seven upregulated cytoplasmic lncRNAs in the MCAO group for further analysis. Additionally, four lncRNAs with no significant differential expression and six downregulated lncRNAs were randomly chosen for quantitative detection. The results indicated that the expression trends of these lncRNAs were consistent with the high-throughput sequencing data, further validating the reliability of our lncRNA sequencing results.

In terms of miRNAs, 59.59% of the reads in the MCAO group and 69.21% in the sham group were successfully mapped to the rat reference genome. In the MCAO and sham groups, we identified 651 differentially expressed miRNAs, with 464 upregulated and 187 downregulated (p < 0.05, |log2FC| > 1). The study revealed that miR-497-5p expression was downregulated after oxygen–glucose deprivation/reperfusion (OGD/R) treatment. The downregulation of miR-497-5p exacerbated neuronal injury in an ischemic stroke model, and this effect was mediated through the negative regulation of the BDNF/TrkB/PI3K/Akt signaling pathway [47]. Notably, miR-370-3p expression decreases after ischemia/reperfusion injury and has antiangiogenic effects [48]. A study by Yang et al. (2022) demonstrated that after ischemic stroke, miR-139-3p is significantly upregulated. CircUSP36 acts as a "sponge" for miR-139-3p, inhibiting its activity, which in turn upregulates the expression of SMAD3 and Bcl2, thereby reducing neuronal apoptosis [49]. Research has shown that miR-383-3p is upregulated after ICH, especially in microglia-derived exosomes. This increase in miR-383-3p levels contributes to the worsening of neuronal damage by promoting necroptosis through the inhibition of ATF4 [50]. These studies demonstrated that miR-497-5p and miR-370-3p are downregulated following ischemic stroke, which aligns with our sequencing results. However, miR-139-3p is upregulated postischemia, which contrasts with our sequencing data. Possible reasons for these discrepancies may include variations in ischemic duration, reperfusion time, and injury severity, which can significantly influence miRNA expression. Furthermore, miR-383-3p is upregulated following intracerebral hemorrhage (ICH), which contradicts our sequencing results. This discrepancy may arise from variations in model establishment methodologies. Differences in animal models and experimental techniques may result in divergent miRNA expression patterns under specific conditions. Given the complex regulatory network of miRNAs, investigating their dynamic changes at various stages and evaluating their interactions with target genes will advance our understanding of the specific roles and mechanisms of miRNAs in ischemic stroke.

Using miRDB, we predicted the DEmRNAs that could be adsorbed by these 25 cytoplasmic lncRNAs and exhibited differential expression relative to the sham group. miRDB is a widely recognized online tool for predicting miRNA–lncRNA interactions. Furthermore, high-throughput sequencing identified 2,608 differentially expressed mRNAs, including 1,133 upregulated and 1,475 downregulated mRNAs (|log2FC| > 2, FDR < 0.01). We employed miRanda and TargetScan to predict the mRNAs targeted by these DEmRNAs, further refining the selection to those mRNAs demonstrating significant differential expression relative to the sham group. These two tools are commonly used for predicting miRNA–target gene interactions, and their combined use enhances the reliability and precision of miRNA target predictions. According to the ceRNA theory, lncRNAs and miRNAs typically exhibit opposite expression patterns, whereas lncRNAs and mRNAs show consistent expression patterns. Ultimately, we successfully constructed a ceRNA regulatory network comprising 12 lncRNAs, 16 miRNAs,

and 191 mRNAs. Among the 167 known protein–coding interactions, we identified the top 20 hub genes with the highest scores. Using 11 of these hub genes, which have been reported to be associated with ischemic stroke (IS), we constructed the core subnetwork of the ceRNA network. Through quantitative analysis, we identified 4 lncRNAs with upregulated expression, 4 miRNAs with downregulated expression, and 11 mRNAs with upregulated expression compared with those in the sham group. These findings are highly consistent with our bioinformatics analysis, further confirming the reliability of the data.

In the central nervous system (CNS), microglia are the primary source of inflammatory mediators [51]. Interruption of the cerebral blood supply activates microglia, initiating inflammatory pathways and releasing significant amounts of pro-inflammatory mediators such as IL-1β, IL-6, and TNF-α, which lead to acute inflammatory responses and neuronal damage [52]. Inhibiting the inflammatory factors produced by microglia and targeting these signaling molecules can potentially prevent neuronal death following ischemic stroke (IS) [53]. In GO term enrichment analysis, SERPINE1, PTGS2, and IL6R were concurrently enriched in the "Response to cytokine" and "Response to lipopolysaccharide" categories. Cytokines are key regulatory factors of the inflammatory response. Lipopolysaccharides, which are components of the bacterial outer membrane, can induce a robust inflammatory response. In this study, we revealed that upregulated MSTRG.151823.1 may specifically interact with miR-370-3p, thereby modulating multiple inflammation-related genes, including PTGS2 and IL6R. PTGS2 and IL6, along with their receptors, primarily promote the inflammatory response following cerebral ischemia. IL6 binds to IL6R, activating downstream JAK-STAT and MAPK signaling pathways, which in turn promotes the production of inflammatory factors and the infiltration of inflammatory cells, ultimately exacerbating neural damage following cerebral ischemia [54]. During cerebral ischemia, PTGS2 expression is upregulated, which coincides with the infiltration of inflammatory cells into the damaged brain [55]. SERPINE1 plays dual roles in the inflammatory response, exhibiting both anti-inflammatory and proinflammatory effects. On the one hand, SERPINE1 can inhibit the migration and phagocytic activity of microglia, thereby exerting an anti-inflammatory effect and mitigating cerebral ischemic damage [56]. Conversely, SERPINE1 can regulate the activity of tPA, leading to the activation of the NF-κB signaling pathway, which in turn exacerbates cerebral ischemic damage [57]. Nevertheless, given the dual role of SERPINE1 in the inflammatory response, its specific effect on cerebral ischemia may be influenced by various factors, including the pathological stage of cerebral ischemia, the inflammatory microenvironment, and the interplay between SERPINE1 and other inflammatory regulatory factors. Thus, further investigations are warranted to elucidate the role of SERPINE1 in the inflammatory response following cerebral ischemia and its underlying regulatory mechanisms.

Apoptosis is the primary pathophysiological mechanism of cell death following stroke, especially in the penumbra region near the ischemic core [58]. Ischemic events in the brain trigger a series of cellular processes that lead to neuronal apoptosis. GO analysis demonstrated that SERPINE1, FURIN, and JAK3 were significantly associated with protease binding terms. Protease binding typically indicates that these gene products regulate the function or activity of proteases, thereby influencing crucial cellular processes, including apoptosis, inflammation, and signal transduction [59]. Therefore, reducing neuronal apoptosis is crucial for developing effective treatments for cerebral ischemia. BAG3 is known as an antiapoptotic protein because it synergizes with Bcl-2. Previous studies have shown that activated autophagy can inhibit apoptosis in myocardial cells after hypoxia/reoxygenation. Xia Liu et al. demonstrated that BAG3 overexpression enhances autophagy and inhibits apoptosis, indicating that autophagy activation and apoptosis inhibition are critical for the neuroprotective role of BAG3 during ischemia/reperfusion and hypoxia/reoxygenation injury [60]. Importantly, SERPINE1 also has a

regulatory function in neuronal apoptosis. SERPINE1 overexpression can counteract the cell apoptosis signal induced by circCNOT6L depletion [61]. Conversely, IL-6R overexpression can diminish the protective role of hsa-miR-21-5p, decrease cell activity, and facilitate cell apoptosis [62]. Our study revealed that BAG3, SERPINE1 and IL-6R are regulated by miR-370-3p. Through lncRNA and miRNA target prediction, we determined that MSTRG.151823.1 can effectively bind to miR-370-3p. These findings suggest that MSTRG.151823.1 modulates apoptosis by regulating BAG3, SERPINE1 and IL-6R through miR-370-3p.

Angiogenesis, the formation of new blood vessels, is induced by the budding of endothelial cells (ECs) from preexisting vessels and occurs under various physiological (e.g., reproductive) and pathological (e.g., ischemic stroke) conditions [63]. Under physiological conditions, proangiogenic and antiangiogenic factors maintain a dynamic equilibrium. Disruption of this balance can result in inadequate angiogenesis, leading to delayed wound healing, such as in cases of stroke. During stroke, hypoxic tissues respond to oxygen deprivation by releasing vascular endothelial growth factor (VEGF), which activates local endothelial cells (ECs) and initiates angiogenic responses to restore the blood supply and normalize oxygen levels. GO analysis indicates that SERPINE1, ITGA2B, FLNA, PTGS2, and ENG are significantly enriched in angiogenesis-related pathways. Additionally, KEGG analysis reveals that the PI3K-Akt signaling pathway is the most significantly enriched pathway among the 20 differentially expressed hub genes. JAK3 and IL6R exhibit significant enrichment within this pathway. This pathway regulates myocardial cell survival, angiogenesis, and the physiological balance of the cardiovascular system [64]. The upregulation of MSTRG.151823.1 sequesters miR-370-3p, leading to the upregulation of Flna, which promotes angiogenesis. However, it also upregulates Ptgs2 and Serpine1, thereby inhibiting angiogenesis. The contradictory roles of MSTRG.151823.1 in promoting and inhibiting angiogenesis may be due to the complex regulatory environment in the brain after stroke. During a stroke, various signaling pathways are activated to either promote recovery or exacerbate damage. The upregulation of Flna by MSTRG.151823.1 aids in vascular repair and angiogenesis, which is crucial for recovery. However, the simultaneous upregulation of Ptgs2 and Serpine1, which are involved in inflammatory responses and the inhibition of angiogenesis, indicates a regulatory feedback mechanism aimed at balancing vascular growth and inflammation to prevent excessive angiogenesis that could lead to further complications. This dual regulatory mechanism reflects the body's attempt to modulate vascular and inflammatory responses to optimize recovery and minimize damage in the stroke-affected brain.

In our study, we revealed that IL6R promotes both inflammation and apoptosis, PTGS2 drives inflammation while suppressing angiogenesis, and LGALS3 enhances inflammation and activates microglia. These findings underscore the central role of inflammation in the pathophysiology of ischemic stroke. Although apoptosis and angiogenesis contribute significantly to tissue repair and pathological responses, inflammation remains the core mechanism driving the entire process. Effective regulation of the inflammatory response can mitigate abnormal changes in apoptosis and angiogenesis while also modulating the overall pathological response, ultimately enhancing neuroprotection and functional recovery. Ischemic stroke is a multifaceted disease with cascade responses to ischemia that extend beyond a linear process and often involves parallel mechanisms and interactions with various factors [65–67]. Key events such as angiogenesis, axonal regeneration, poststroke inflammation, microglial activation, BBB disruption, apoptosis, autophagy, ferroptosis, and platelet reduction are interrelated. These processes can have both neuroprotective and neurodestructive effects, thereby influencing disease prognosis and outcome.

Moreover, the regulation of lncRNAs lacks complete specificity (one-to-many, many-to-one), highlighting that complex diseases such as stroke result from cumulative biological

events. These events involve various functional genes whose expression, activity, and localization interact in a dynamic and reciprocal manner. The influences of these interactions accumulate and counterbalance each other across intensity, duration, and spatial distribution, thereby modulating the progression and prognosis of stroke. Despite advances in our understanding of lncRNAs, this study has limitations. For example, in predicting miRNA-targeted mRNAs, we retained only those mRNAs that demonstrated a countertrend toward miRNA expression, which relies on changes at the mRNA level and may not be exhaustive. This approach may overlook mRNAs that do not show significant changes at the mRNA level but have notable changes at the protein level. Additionally, RT–qPCR validated only the mRNA levels of key genes without assessing their protein levels. And it is crucial to acknowledge that these findings are primarily based on bioinformatics predictions and will require experimental validation to establish their functions in the disease process. Given the dynamic complexity of stroke development, future research should explore network mechanisms involving multiple molecules, pathways, and various regulatory levels to provide a more comprehensive understanding of this disease.

## 5 Conclusions

In this study, we employed the Middle Cerebral Artery Occlusion (MCAO) rat model to investigate the pathophysiological mechanisms underlying ischemic stroke. Using high-throughput sequencing, we generated comprehensive expression profiles of long non-coding RNAs (lncRNAs), microRNAs (miRNAs), and mRNAs. Differential expression analysis, combined with bioinformatics tools such as miRanda, TargetScan, and miRDB, enabled us to predict miRNA-lncRNA and miRNA-mRNA interactions, culminating in the identification of several key regulatory networks. Our analysis revealed 12 lncRNAs, 16 miRNAs, and 191 mRNAs that constitute a core ceRNA (competing endogenous RNA) network, with 25 of these lncRNAs localized in the cytoplasm. We identified 11 hub genes associated with ischemic stroke within this network. We further delineated a core subnetwork encompassing 4 lncRNAs, 4 miRNAs, and 11 mRNAs. Among the significant regulatory pathways,MSTRG.151823.1-miR-370-3p-Ptgs2/Il6r, MSTRG.29312.2/MSTRG.243513.1-miR-139-3p-Jak3, and MSTRG.82425.1-miR-351-5p-Lgals3 may significantly influence the inflammatory response during ischemia-reperfusion injury. These findings provide insights into the molecular mechanisms of ischemic stroke, highlighting the complexity of lncRNA-miRNA-mRNA interactions. The identification of these key regulatory networks offers potential targets for therapeutic intervention aimed at improving outcomes in ischemic stroke patients.

## Supporting information

**S1 Fig. S1A; Classification map of 31183 lncRNAs obtained by sequencing.** From left to right, they are intergenic lncRNA (red); intronic lncRNA (green); antisense lncRNA (blue); and sense lncRNA (purple). S1B: Circular diagram of lncRNA distribution on chromosomes. The outermost layer is the chromosome ring of the genome, and from outside to inside are sense lncRNA (green), intergenic region lncRNA (red), intronic lncRNA (blue), and antisense lncRNA (gray).
(DOCX)

**S1 Table. Primer sequence.**
(DOCX)

**S2 Table. Summary of the mapping data from the cortex tissue.**
(DOCX)

**S3 Table. 31183 lncRNAs classification.**
(XLS)

**S4 Table. Up-regulated or down-regulated(DEL).**
(XLS)

**S5 Table. Cunts, FPKM and FDR of DEL in 10 sequencing samples.**
(XLS)

**S6 Table. Screening of core DEL.**
(XLS)

**S7 Table. The subcellular localization of the 33 lncRNAs.**
(XLSX)

**S8 Table. miRNA Data evaluation statistics.**
(DOCX)

**S9 Table. The 651 differentially expressed miRNAs.**
(XLSX)

**S10 Table. The 2608 differentially expressed mRNAs.**
(XLSX)

**S11 Table. LncRNA-miRNA.**
(XLSX)

**S12 Table. miRNA-mRNA.**
(XLSX)

**S13 Table. lncRNA-miRNA-mRNA.**
(XLSX)

**S14 Table. Protein–protein interaction network assessment.**
(XLSX)

**S15 Table. Biological process enrich.**
(XLSX)

**S16 Table. Cellular component enrich.**
(XLSX)

**S17 Table. Molecular function enrich.**
(XLSX)

**S18 Table. KEGG pathway enrich.**
(XLSX)

**S19 Table. The count data from the microRNA expression analysis.**
(XLSX)

## Acknowledgments

## Declarations

All animal procedures were conducted following the guidelines established by the Institutional Animal Care and Use Committee (IACUC) of the First Affiliated Hospital of Shihezi

University School of Medicine. The approval number is A2022-037-01. Animal care and use strictly followed the recommendations outlined in the Chinese National Regulations for Experimental Animal Management.

## Author Contributions

**Conceptualization:** Meimei Xu.

**Data curation:** Shan Yuan.

**Formal analysis:** Meimei Xu, Guangze Hu.

**Funding acquisition:** Rui Gao.

**Investigation:** Shan Yuan, Guangze Hu.

**Methodology:** Xing Luo, Zhe He, Xinyuan Yang.

**Project administration:** Mengsi Xu.

**Resources:** Mengsi Xu.

**Software:** Zhe He, Xinyuan Yang.

**Supervision:** Xing Luo.

**Validation:** Meimei Xu.

**Writing – original draft:** Meimei Xu.

**Writing – review & editing:** Rui Gao.

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
