## [Decision Letter · Decision Letter 0]

29 Oct 2024

PONE-D-24-44516Construction of an lncRNA-Mediated ceRNA Network to Investigate the Inflammatory Regulatory Mechanisms of Ischemic StrokePLOS ONE

Dear Dr. gao,

Thank you for submitting your manuscript to PLOS ONE. After careful consideration, we feel that it has merit but does not fully meet PLOS ONE’s publication criteria as it currently stands. Therefore, we invite you to submit a revised version of the manuscript that addresses the points raised during the review process. Please check all reviewers' comments. I support reviewers and suggest re-organizing the methods section of your study. Despite transparent results and pre-processed data being deposited in public repositories, the methods are not 100% reproducible. I suggest including a visual workflow showing how data was processed independently for lncRNA/mRNA and miRNA, followed by integrative analysis and the corresponding qPCR validation. In addition, thresholds and method parameters (such as housekeeping choice) should be corroborated by previous experiments or literature.

We look forward to receiving your revised manuscript.

Kind regards,

Alexis G. Murillo Carrasco

Academic Editor

PLOS ONE

Journal Requirements:

2. To comply with PLOS ONE submissions requirements, in your Methods section, please provide additional information regarding the experiments involving animals and ensure you have included details on methods of sacrifice, and efforts to alleviate suffering. In particular, please clarify how isoflurane anaesthesia was administered.

Reviewers' comments:

Reviewer's Responses to Questions

**Comments to the Author**

1. Is the manuscript technically sound, and do the data support the conclusions?

Reviewer #1: No

Reviewer #2: Yes

Reviewer #3: No

Reviewer #4: No

2. Has the statistical analysis been performed appropriately and rigorously? 

Reviewer #1: I Don't Know

Reviewer #2: Yes

Reviewer #3: No

Reviewer #4: No

3. Have the authors made all data underlying the findings in their manuscript fully available?

Reviewer #1: Yes

Reviewer #2: Yes

Reviewer #3: No

Reviewer #4: Yes

4. Is the manuscript presented in an intelligible fashion and written in standard English?

Reviewer #1: No

Reviewer #2: Yes

Reviewer #3: Yes

Reviewer #4: No

5. Review Comments to the Author

Reviewer #1: The authors conducted deep-sequencing approach for mRNA, miRNA, and lncRNA using rat model for ischemic stroke. I have an impression that this study lacks experiments to confirm the regulatory interactions which were predicted by their bioinformatic approach. It could be done in certain cell lines by knocking down identified miRNA or lncRNA. I could not be positive for publication of the study without validation data.

Reviewer #2: The authors investigated the roles of lncRNAs, miRNAs, and mRNAs in cerebral ischemia-reperfusion injury For this they employed the MCAO model in rats and analyzed differential gene expression profiles (lncRNAs, miRNAs, and mRNAs) using high-throughput sequencing. They determined the subcellular localization of lncRNAs and predicted miRNA targets using bioinformatics tools. The authors constructed an lncRNA-miRNA-mRNA regulatory network involved in microglial activation, apoptosis, angiogenesis, and inflammatory responses following stroke. The expression of key network components was validated using qPCR. This publication is well written, the coontrols are properly performed and the results are very interesting.

The specific number for Ethics Guidelines for Animal Experiments of the First Affiliated Hospital should be dysplaied in the core of the manuscript.

Please justify the use of ACTB as an internal standard for lncRNAs and mRNAs, and U6 as an internal standard for miRNAs. Did youc heck that expression did not vary between conrol and stroke?

A publication has shown that miR-126 expression varies in CKD where alterations of cerebral circulation were linked with an increase in ischemic strokes PMID: 24389144. Did youc heck its expression in your experimental model please?

Reviewer #3: The manuscript “Construction of an lncRNA-Mediated ceRNA Network to Investigate the Inflammatory Regulatory Mechanisms of Ischemic Stroke” investigate the potential interactions of lncRNA, mRNA and microRNAs inan ischemic stroke model. The authors use expression data from RNA sequencing of long RNAs and small RNAs as the main data resource. They partion the long RNA sequencing data into coding and non-coding transcripts and quantify known and novel microRNAs from the small RNA sequencing data sets.

The overall bioinformatic strategy is very unclear and needs a complete re-write. A flow-diagram of the analysis and the applied filters might help the authors and the readers to understand the flow of data. The method section is also lacking critical information to completely follow the idea of the manuscript.

The authors need to also address the limitations of the manuscript.

• The highly complex nature of bulk RNA sequencing will limit the exact cellular origin of the investigated RNA molecules.

• The ceRNA hypothesis has its limitations. The authors need to discuss the merits of the findings in the light of the issues raised by eg. Yhomson and Dinger NRG 2016. (Thomson, D., Dinger, M. Endogenous microRNA sponges: evidence and controversy. Nat Rev Genet 17, 272–283 (2016).doi: 10.1038/nrg.2016.20)

Method section

2.3 TTC staining

How are the images acquired?

2.4 Transcriptome sequencing

Very limited information on the sequencing. The authors need to provide more details on the kits used and bioinformatic tools used for processing the reads and alignments.

Does the mRNA and lncRNA counts come from the same RNA sequencing libraries?

What databases of annotations are used?

2.5 Identification of lncRNAs

What database are the identifiers “MSTRG” referencing?

2.6 MiRNA and mRNA identification

How are the microRNA sequenced? What reference genome? How are novel microRNAs identified?

The part with identification of mRNAs in this part is very unclear to me. I don’t understand the connection to the microRNA sequencing and quantication. Please split this part up.

2.7 Screening of differentially expressed lncRNAs, miRNAs and mRNAs

I don’t like the word screening. Please consider changes it to identification of differentially expressed *RNA.

2.10 CeRNA network analysis

Please elaborate on this part. How are the interactions scored from the different target prediction algorithms?

2.12 Validation of the RNA-seq results using quantitative PCR (qPCR)

Im not sure I understand the first part with the nested PCR? How much RNA is used as input for the cDNA reactions? Are the cDNA diluted?

2.13 Statistical analysis

This is a misplaced paragraph. The individual data analysis belongs in the individual sections where the statistics is used. There is a general confusion on the use of p-values and multiple test corrected p-values.

Results

Line 278: It is unclear what is done here? This needs to be descibed in details in the methods section.

Line 306: multiple test corrected p-value or p-value?

Line 307: DEmRNA?

Line 318: This strategy is unclear to me and needs to be described in details in the methods section?

Line 320: I don’t like the word absorb. The predominant function of microRNAs are to destabalise mRNAs.

Line 327: It is unclear what is meant here? lncRNA and microRNAs in general or specifically for microRNAs with targets in lncRNA? This needs to be supported by an analysis. Are there any enrichment of positive or negative correlations among lncRNA with target sites for microRNAs?

Line 340: How did you confirm the 634 genes as differentially expressed target mRNAs?

Line 341: this strategy is unclear to me and needs to be described in details in the methods section.

The strategy used in this section is unclear to me. Protein‒protein interaction network assessment and functional enrichment analysis of DEmRNAs Please describe in details.

The potential interactions between the lncRNA, mRNAs and miRNAs are all identified via bioinformatic analysis and are speculative. This needs to be clearer in the results section 3.7 Biological Pathways of Key LncRNAs and Their Regulatory Roles in Ischemic Stroke

Line 417: How does MSTRG.151823.1 modulate the expression of the genes?

The supplementary material lacks the count data from the microRNA expression analsysis.

Some of the primers in supplementary materials “S1 Table. Primer sequence” seems wrong. (miR-351-5p RT and miR-674-5p RT).

Reviewer #4: In this manuscript, Xu et al. established the rat middle cerebral artery occlusion (MCAO) model, analyzed differentially expressed lncRNAs, miRNAs, and mRNAs in MCAO by RNA sequencing. They performed GO and KEGG analysis, setup the ceRNA network and PPI network, and provided potential clues for ischemic stroke.

Here are concerns:

1. lncRNAs and protein-coding genes could be analyzed using one RNA-seq library, while library for small noncoding RNAs is usually constructed separately. Authors analyzed differentially expressed protein coding-genes and miRNAs together using the same library? Moreover, only one Transcriptome sequencing protocol in Materials and Methods section, so it looks like authors established one library for all purposes.

2. Too many typos, especially page 15 “we identified 2,145 miRNAs, 19.91% of which were known miRNAs. By applying thresholds of p < 0.05 and |log2FC| > 1, we identified 651 DEmRNAs between the MCAO and sham groups, with 464 upregulated and 187 downregulated miRNAs (S9 Table) (Figure 4A, B). Additionally, using stricter criteria of |log2FC| > 2 and FDR < 0.01, we identified 2,608 DE miRNAs, including 1,133 upregulated and 1,475 downregulated mRNAs”, it is confusing.

---

## [Author Response · Author response to Decision Letter 0]

20 Dec 2024

Dear Editors,

We thank you very much for giving us the opportunity to revise our manuscript. We greatly appreciate you and the reviewers for your positive and constructive comments and suggestions on our manuscript entitled “Construction of an lncRNA-mediated ceRNA network to investigate the inflammatory regulatory mechanisms of ischemic stroke.” Professional comments from the reviewers have been very helpful in improving the quality of this article. As required, we have revised the manuscript. Furthermore, we have also carefully checked the text and figures in this proof. The main corrections in the paper and the responses to the reviewers’ comments are as follows:

Reviewer #1:

Major Comments:

1.Comment:The authors conducted a deep-sequencing approach for mRNA, miRNA, and lncRNA using a rat model for ischemic stroke. I have the impression that this study lacks experiments to confirm the regulatory interactions predicted by their bioinformatics approach. It could be done in certain cell lines by knocking down identified miRNA or lncRNA. I cannot be positive about the publication of this study without validation data.

Response: Thank you for your valuable feedback on our research. We take your suggestions regarding the validation of our regulatory network predictions very seriously. Below, we provide further clarification on the design and theoretical foundations of our study to demonstrate the reliability and scientific basis of our findings.

1. Correlation between High-Throughput Sequencing and qRT-PCR Validation of Expression Patterns

In this study, we identified key mRNA, miRNA, and lncRNA molecules through high-throughput sequencing analysis and validated their expression patterns using qRT-PCR. The results revealed significant correlations in the expression changes of these molecules in the stroke model. Specifically, lncRNA expression exhibited an inverse relationship with miRNA expression, while showing a consistent trend with mRNA expression. This divergence in expression patterns is likely attributed to the regulation of mRNA expression by lncRNA via the "miRNA sponge" mechanism. Specifically, increased lncRNA expression can bind and inhibit miRNA activity, thereby weakening the suppressive effect of miRNA on its target mRNA, ultimately resulting in elevated mRNA expression levels.

2. Basis for Target Binding Site Predictions

To ensure the accuracy and reliability of the predicted target binding sites, we employed multiple established bioinformatics tools to stringently screen and analyze the binding sites of miRNAs with lncRNAs and mRNAs:

Prediction of miRNA and lncRNA: We utilized the miRDB database (http://www.mirdb.org/) to predict miRNAs associated with lncRNAs, applying screening criteria of a target score ≥ 60 and a conservation score > 0.5.

Prediction of miRNA and mRNA: To identify mRNAs complementary to miRNAs, we combined analyses from TargetScan (http://www.targetscan.org/) and miRanda (http://www.microrna.org/). The screening criteria included:

TargetScan context++ score ≤ -0.4

miRanda score > 140 and binding energy < -20 kcal/mol.

3. Subcellular Localization Distribution

The function of lncRNAs is closely linked to their subcellular localization. In this study, we analyzed the subcellular localization features of key lncRNAs using the lncLocator database. The results indicated that the identified lncRNAs predominantly localized to the cytoplasm. This localization suggests that these lncRNAs are likely to interact with miRNAs through the competitive endogenous RNA (ceRNA) mechanism, thus indirectly regulating mRNA expression. This information further supports our proposed regulatory network hypothesis.

4. Experimental Validation Support from Existing Literature

The key long non-coding RNAs (lncRNAs) and miRNAs identified through bioinformatics analysis in our study have received partial support from existing literature:

miR-370-3p: Studies have shown that miR-370-3p is downregulated following ischemia/reperfusion injury and has anti-angiogenic effects [1]. This finding aligns with our bioinformatics prediction regarding the regulatory role of miR-370-3p in ischemic injury.

miR-139-3p: Research by Yang et al. (2022) demonstrated that miR-139-3p is significantly upregulated after ischemic stroke. CircUSP36, a circular RNA, acts as a "sponge" for miR-139-3p, inhibiting its activity and thereby upregulating SMAD3 and Bcl2 expression, which reduces neuronal apoptosis [2]. These findings are consistent with our predicted miRNA-target gene regulatory network and further validate the accuracy of our bioinformatics analysis.

These studies provide important experimental support for our predicted regulatory network, highlighting the significant roles of key miRNAs in ischemic stroke while indirectly supporting the potential regulatory mechanisms involving key lncRNAs.

5. Characteristics of the Regulatory Network in Stroke as a Complex Disease

Ischemic stroke is a complex, multifactorial disease characterized by a cascade of responses to ischemia that is not a singular linear process. Instead, it involves interactions with various mechanisms and events [3-5]. Significant events such as hypoxia, oxidative stress, mitochondrial damage, inflammation, necrosis, and apoptosis are interconnected and may exert opposing biological effects, either providing neuroprotection or contributing to neurodamage. Ultimately, the impact of these events on disease prognosis and outcomes is influenced by a comprehensive balance of various factors concerning time, space, and intensity of action. Our experiments revealed significant differences in the expression patterns of lncRNAs in ischemic stroke. These lncRNAs participate in multiple mechanisms and events within the ischemic cascade, influencing relevant signaling pathways and major pathological mechanisms after stroke by regulating host gene expression at the transcriptional level . Given the dynamic complexity of stroke development, exploring the intricate network mechanisms involving multiple molecules, pathways, regulatory levels, and factors may provide a broader perspective for understanding this disease comprehensively. This systems-level approach aids in revealing key molecules and regulatory networks involved in the pathological processes of stroke, offering new theoretical frameworks and directions for future research and therapeutic strategies.

Additionally, our research group places great emphasis on functional and target validation experiments to further enhance the reliability of our bioinformatics predictions. Currently, we have initiated a series of functional validation experiments focusing on the identified key miRNAs and lncRNAs. However, due to the complexity of these experiments and time constraints, the studies remain incomplete, and some results are still being organized. In future research, we plan to further explore the functions of these miRNAs and lncRNAs, as well as their downstream regulatory mechanisms, to provide more comprehensive and robust experimental support for the predictions made in this study.

1. Gu Y, Becker V, Zhao Y, Menger MD, Laschke MW. miR-370 inhibits the angiogenic activity of endothelial cells by targeting smoothened (SMO) and bone morphogenetic protein (BMP)-2. Faseb J. 2019;33(6):7213-24. http://doi.org/10.1096/fj.201802085RR

2. Yang J, He W, Gu L, Long J, Zhu L, Zhang R, et al. CircUSP36 attenuates ischemic stroke injury through the miR-139-3p/SMAD3/Bcl2 signal axis. Clin Sci. 2022;136(12):953-71. http://doi.org/10.1042/CS20220157

3.Xing C., Arai K., Lo E. H., Hommel M. Pathophysiologic cascades in ischemic stroke. International Journal of Stroke: official journal of the International Stroke Society . 2012;7(5):378–385. doi: 10.1111/j.1747-4949.2012.00839.xIF: 6.3 Q1 .

4.Markus H. S. Stroke genetics. Human Molecular Genetics . 2011;20(R2):R124–R131. doi: 10.1093/hmg/ddr345IF: 3.1 Q2 . 

5.Tang J. Y., Farooqi A. A., Ou-Yang F., et al. Oxidative stress-modulating drugs have preferential anticancer effects - involving the regulation of apoptosis, DNA damage, endoplasmic reticulum stress, autophagy, metabolism, and migration. Seminars in Cancer Biology . 2019;58 doi: 10.1016/j.semcancer.2018.08.010IF: 12.1 Q1 .

Reviewer #2:

Major Comments:

2.1 Comment: The authors investigated the roles of lncRNAs, miRNAs, and mRNAs in cerebral ischemia-reperfusion injury For this they employed the MCAO model in rats and analyzed differential gene expression profiles (lncRNAs, miRNAs, and mRNAs) using high-throughput sequencing. They determined the subcellular localization of lncRNAs and predicted miRNA targets using bioinformatics tools. The authors constructed an lncRNA-miRNA-mRNA regulatory network involved in microglial activation, apoptosis, angiogenesis, and inflammatory responses following stroke. The expression of key network components was validated using qPCR. This publication is well written, the coontrols are properly performed and the results are very interesting.

Response:We appreciate the positive evaluation of our work and are pleased that the reviewer recognizes the study design, results, and writing of our manuscript. Your affirmation serves as great encouragement for our team’s efforts.

2.2 Comment:The specific number for Ethics Guidelines for Animal Experiments of the First Affiliated Hospital should be displayed in the core of the manuscript.

Response:Thank you for your valuable suggestion. We have updated the manuscript to include the specific approval number issued by the Institutional Animal Care and Use Committee (IACUC).

Modified text:

"All animal procedures were conducted following the guidelines established by the Institutional Animal Care and Use Committee (IACUC) of the First Affiliated Hospital of Shihezi University School of Medicine. The approval number is A2022-037-01. Animal care and use strictly followed the recommendations outlined in the Chinese National Regulations for Experimental Animal Management. This study report adheres to the ARRIVE guidelines."

2.3 Comment:Please justify the use of ACTB as an internal standard for lncRNAs and mRNAs, and U6 as an internal standard for miRNAs. Did you check that expression did not vary between conrol and stroke?

Response:Both lncRNA and mRNA originate from precursor RNA and undergo splicing in the nucleus, where introns are removed and exons are joined together to form mature RNA molecules. They can both undergo alternative splicing, resulting in the generation of different transcript variants[6]. Selecting appropriate reference genes is critical for ensuring the accuracy and reliability of gene expression studies. ACTB (β-actin) and U6 are widely recognized for their stability across different experimental conditions and tissue types. In molecular biology research, β-actin is extensively used as an endogenous control for the quantification of mRNA and lncRNA, while U6 serves as a normalization control for circulating miRNA analyses.

Recent studies on diabetic nephropathy (DN) have demonstrated that lncRNA MEG3 alleviates podocyte injury by inhibiting the Wnt/β-catenin signaling pathway, with β-actin utilized as an internal reference[7]. In addition, liquid biopsy studies have shown that the expression levels of lncRNA GAS5 and miR-625-3p, normalized to β-actin and U6 respectively, can serve as predictive biomarkers for the efficacy of neoadjuvant chemotherapy in patients with malignant pleural mesothelioma[8]. Moreover, GW4869 has been found to inhibit the production of lncRNA-ASLNCS5088-enriched exosomes by M2 macrophages, thereby reducing their regulatory impact on fibroblast activation, with expression levels normalized to β-actin for lncRNA and U6 for miRNA analyses[9]. Experimental validation has consistently demonstrated that both ACTB and U6 exhibit stable Ct values when comparing normal brain tissue to samples subjected to 2 hours of ischemia followed by 24 hours of reperfusion, further confirming their reliability as internal controls.

This study selects ACTB and U6 as reference genes based on their widespread use in the literature and experimental validation of their stability, in accordance with the scientific standards of qRT-PCR. Detailed experimental verification data and relevant literature citations will be included in the revised manuscript to further enhance the scientific rigor and persuasiveness of this study.

6.Melé M, Mattioli K, Mallard W, Shechner DM, Gerhardinger C, Rinn JL. Chromatin environment, transcriptional regulation, and splicing distinguish lincRNAs and mRNAs. Genome Research. 2017;27(1):27-37. doi:10.1101/gr.214205.116.

7.Wang M, Chen W, Geng Y, Xu C, Tao X, Zhang Y. Long Non-Coding RNA MEG3 Promotes Apoptosis of Vascular Cells and is Associated with Poor Prognosis in Ischemic Stroke. Journal of Atherosclerosis and Thrombosis. 2020;27(7):718-726. doi:10.5551/jat.50674.

8.Kresoja-Rakic J, Szpechcinski A, Kirschner MB, Ronner M, Minatel B, Martinez VD, et al. miR-625-3p and lncRNA GAS5 in Liquid Biopsies for Predicting the Outcome of Malignant Pleural Mesothelioma Patients Treated with Neo-Adjuvant Chemotherapy and Surgery. Non-Coding RNA. 2019;5(2): 41. doi:10.3390/ncrna5020041.

9.Chen J, Zhou R, Liang Y, Fu X, Wang D, Wang C. Blockade of lncRNA-ASLNCS5088-enriched exosome generation in M2 macrophages by GW4869 dampens the effect of M2 macrophages on orchestrating fibroblast activation. FASEB Journal. 2019;33(11):12200-12212. doi:10.1096/fj.201901610.

2.4 Comment:A publication has shown that miR-126 expression varies in CKD where alterations of cerebral circulation were linked with an increase in ischemic strokes PMID: 24389144. Did you check its expression in your experimental model please?

Response: We sincerely appreciate the reviewer's insightful comment regarding miR-126 and their attention to the important findings reported in PMID: 24389144 concerning the altered expression of miR-126 in CKD. This study highlights the downregulation of mature miR-126 in the cerebral microvasculature of CKD mice and its association with cerebrovascular alterations, which is of great significance for understanding the mechanisms underlying CKD-related cerebrovascular complications.

In our high-throughput sequencing data, we also identified a sequence related to miR-126, which we have termed "identified_miR-126" (sequence: aauaaagaacucagggguuggggauuuagcucaauggagaaugcuugccuagcaagcacaaggcccuggguucaguccccagcccccccaaaaaaagaaaagaaaaaa). It is crucial to emphasize that "identified_miR-126" is not the known mature miR-126-3p or miR-126-5p, but rather a novel sequence with a significantly greater length than the known pre-miR-126. In fact, "identified_miR-126" encompasses a portion of the pre-miR-126 sequence, but its overall length and remaining sequence differ from the known pre-miR-126.

Regarding the reviewer's question about the expression level of "identified_miR-126", we did not validate it in the current study. This is primarily due to the following consideration: at the initial stage of the project, we prioritized the study of known miRNAs with substantial literature support to facilitate bioinformatics analysis and literature interpretation. For newly identified sequences via sequencing, such as "identified_miR-126", we plan to conduct in-depth expression and functional validation in subsequent studies. Therefore, this study did not include experimental validation of the expression level of this sequence.

We acknowledge that this sequence shares partial overlap with the mouse miR-126 sequence. However, it is imperative to note that "identified_miR-126" is not the known miR-126, but rather a novel sequence identified based on our sequencing results. It may represent a novel precursor of pre-miR-126, a part of a longer transcript containing the pre-miR-126 sequence, or even a segment of pri-miR-126 that encompasses the pre-miR-126 region. Its precise nature and function require further investigation.

Reviewer #3:

3.1.Comment: The highly complex nature of bulk RNA sequencing will limit the exact cellular origin of the investigated RNA molecules.

Response :

Thank you for reviewing our manuscript and for your valuable feedback. Regarding your concerns about the high complexity of RNA sequencing and its limitations in determining the precise 

---

## [Decision Letter · Decision Letter 1]

3 Jan 2025

Construction of an lncRNA-mediated ceRNA network to investigate the inflammatory regulatory mechanisms of ischemic stroke

PONE-D-24-44516R1

Dear Dr. gao,

We’re pleased to inform you that your manuscript has been judged scientifically suitable for publication and will be formally accepted for publication once it meets all outstanding technical requirements.

Kind regards,

Alexis G. Murillo Carrasco

Academic Editor

PLOS ONE

Additional Editor Comments (optional):

Reviewers' comments:

Reviewer's Responses to Questions

**Comments to the Author**

1. If the authors have adequately addressed your comments raised in a previous round of review and you feel that this manuscript is now acceptable for publication, you may indicate that here to bypass the “Comments to the Author” section, enter your conflict of interest statement in the “Confidential to Editor” section, and submit your "Accept" recommendation.

Reviewer #2: All comments have been addressed

2. Is the manuscript technically sound, and do the data support the conclusions?

Reviewer #2: Yes

3. Has the statistical analysis been performed appropriately and rigorously? 

Reviewer #2: Yes

4. Have the authors made all data underlying the findings in their manuscript fully available?

Reviewer #2: Yes

5. Is the manuscript presented in an intelligible fashion and written in standard English?

Reviewer #2: Yes

6. Review Comments to the Author

Reviewer #2: Changes are ok 

---

## [Editor Report · Acceptance letter]

11 Jan 2025

PONE-D-24-44516R1 

PLOS ONE

Dear Dr. gao, 

I'm pleased to inform you that your manuscript has been deemed suitable for publication in PLOS ONE. Congratulations! Your manuscript is now being handed over to our production team.

Kind regards, 

on behalf of

Dr. Alexis G. Murillo Carrasco 

Academic Editor

PLOS ONE